# SciArena: An Open Evaluation Platform for Non-Verifiable Scientific Literature-Grounded Tasks

Yilun Zhao[*][Y]  Kaiyan Zhang[*][Y]  Tiansheng Hu[*][NY]  Sihong Wu[*][Y]  Ronan Le Bras[*][A]
Charles McGrady[*][A]  Taira Anderson[A]  Jonathan Bragg[A]  Joseph Chee Chang[A]
Jesse Dodge[A]  Matt Latzke[A]  Yixin Liu[Y]  Xiangru Tang[Y]  Zihang Wang[NY]
Chen Zhao[NY]  Hannaneh Hajishirzi[*][A][W]  Doug Downey[*][A][NU]  Arman Cohan[*][Y]

[Y]Yale University    [NY]New York University    [A]Allen Institute for AI
[W]University of Washington    [NU]Northwestern University

{yilun.zhao, arman.cohan}@yale.edu, dougd@allenai.org

Platform: sciarena.allen.ai
Data: yale-nlp/SciArena    Code: yale-nlp/SciArena

## Abstract

We present SciArena, an open and collaborative platform for evaluating foundation models on scientific literature-grounded tasks. Unlike traditional benchmarks for scientific literature understanding and synthesis, SciArena engages the research community directly, following the Chatbot Arena evaluation approach of community voting on model comparisons. By leveraging collective intelligence, SciArena offers a community-driven evaluation of model performance on open-ended scientific tasks that demand literature-grounded, long-form responses. The platform currently supports 47 foundation models and has collected over 20,000 votes from human researchers across diverse scientific domains. Our analysis of the data collected so far confirms its high quality. We discuss the results and insights based on the model ranking leaderboard. To further promote research in building model-based automated evaluation systems for literature tasks, we release SciArena-Eval, a meta-evaluation benchmark based on collected preference data. It measures the accuracy of models in judging answer quality by comparing their pairwise assessments with human votes. Our experiments highlight the benchmark's challenges and emphasize the need for more reliable automated evaluation methods.

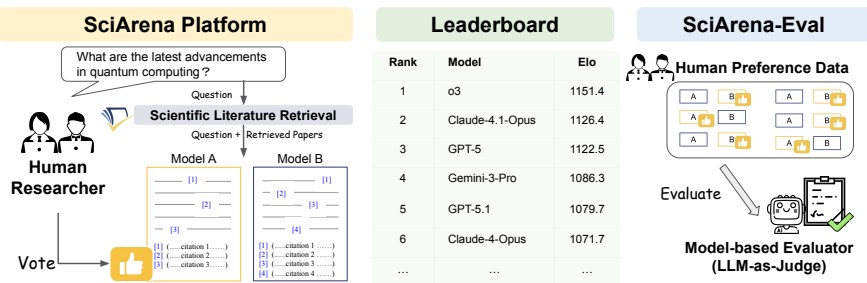

Figure 1: SciArena focuses on evaluating foundation models on scientific literature tasks. It consists of three main components: (1) a platform that collects human researcher preference votes between foundation models; (2) a leaderboard that ranks models using an Elo rating system based on these votes; and (3) the SciArena-Eval benchmark for assessing model-based evaluation systems.

---

[*]Core contributors. The remaining authors are listed alphabetically by last name. The first three authors contributed equally.

39th Conference on Neural Information Processing Systems (NeurIPS 2025) Track on Datasets and Benchmarks.

# 1   Introduction

Scientific literature understanding and synthesis play a pivotal role in uncovering research gaps, guiding methodological innovation, informing practical application, and enabling scientific discovery [67, 78, 85, 12]. However, the exponential growth in scholarly publications poses significant challenges for researchers attempting to maintain comprehensive awareness of developments within their fields. To assist with this challenge, foundation models are increasingly leveraged to help researchers in the discovery, synthesis, and interpretation of scholarly content [55, 19, 69, 7, 89, 77, 87, 81].

At the same time, evaluating the capabilities and limitations of foundation models in open-ended scientific literature-grounded tasks remains challenging. Recent work in evaluation of open-ended instruction-following tasks often relies on LLM-based evaluators [13, 88]. Despite the scalability of LLM-based evaluation methods for evaluating model responses, such evaluators can fail to achieve high alignment with expert annotators across diverse datasets [43, 44]. This is especially the case in science where nuanced, domain-specialized, and knowledge-intensive requirements can be overlooked by LLM-based evaluations [47, 8, 73]. On the other hand, obtaining human expert judgments is widely acknowledged as both time-consuming and expensive, especially in knowledge-intensive domains [8, 68]. Consequently, most expert-annotated benchmarks in science are static, remain limited in scale and quickly become outdated, especially in fast-evolving scientific fields-of-study.

To address these challenges, we introduce SciArena, a collaborative and open platform inspired by ChatBot Arena [14] that harnesses the collective expertise of the scientific community. SciArena serves as an interactive evaluation platform for scientific literature-grounded tasks where users can submit questions related to up-to-date research, view side-by-side literature-grounded and long-form responses generated by foundation models, and vote for their preferred output. Unlike general-purpose tasks targeted by existing arena platforms, scientific literature tasks demand a high degree of domain expertise and precise literature retrieval. To meet these specialized requirements, we implement a multi-stage retrieval pipeline adapted from Ai2's Scholar QA system [69], which includes query decomposition, passage retrieval, and re-ranking. The retrieved paper contexts, combined with the user's question, are provided to two foundation models, each generating long-form literature-based responses with citations. Users then evaluate these outputs and vote for the one that best satisfies their information need.

Over the first eight months of SciArena operation, we have collected over 20,000 votes from human researchers across diverse scientific fields. We apply rigorous quality control and conduct detailed analyses to ensure the reliability and integrity of the human preference data. Using this data, we construct the SciArena leaderboard, identifying the state-of-the-art models as o3, Claude-4.1-series, and GPT-5 models. We further perform in-depth analyses, including breakdowns of model performance across scientific disciplines and question categories, head-to-head win rates between models, and qualitative assessments of model failure cases, highlighting key insights for advancing foundation models in scientific literature tasks.

Despite the limitations of automated evaluation methods, systematically studying how well model-based evaluators can approximate human experts judgments on long-form, literature-based responses remains underexplored. Understanding this alignment gap could reveal both the current boundaries of automated evaluations in science and inspire targeted improvements in evaluation methodologies. To this end, we introduce the SciArena-Eval meta-evaluation benchmark, constructed using user voting data, to facilitate the development of model-based automated evaluation systems. SciArena-Eval assesses how closely model-based evaluators align with human preferences, providing insight into their evaluation capabilities. Our findings reveal that even the best-performing evaluation system, o3 with pairwise comparison, achieves only 65.1% accuracy compared with human preference. This underscores the need for more robust evaluation approaches.

Figure 1 presents an overview of this study. We summarize our main contributions as follows:

- We introduce SciArena, the first open evaluation platform for ranking foundation models in non-verifiable scientific literature-grounded tasks, based on preferences from human researchers.

- We collect over 20,000 votes from researchers across different scientific domains. Our quality assessment and case studies demonstrate the high quality of collected data and the strong reliability of SciArena. We publicly release this human preference data to support future research.

- We develop the SciArena leaderboard using collected human preference data, providing researchers and developers with a comprehensive understanding of state-of-the-art models in scientific literature tasks. Our in-depth analyses highlight key insights for advancing foundation models in this domain.
- We construct SciArena-Eval, the first benchmark designed to assess model-based evaluators in judging citation-attributed responses to user questions. Our experiments show that existing methods perform poorly on SciArena-Eval, highlighting the need for more robust evaluation approaches.

## 2   Related Work

**Foundation Models for Scientific Literature-Grounded Tasks.** Understanding and synthesizing scientific literature is a cornerstone of research progress and innovation. Recent advances have introduced foundation model-based systems powered by retrieval-augmented generation (RAG) frameworks [8, 69, 17, 65, 25, 61, 83, 55, 19] that leverage sophisticated retrieval and generation mechanisms to support literature reviews and accelerate knowledge synthesis. However, many existing systems (*e.g.,* OpenAI's Deep Research [55]) that are accessible without local deployment are primarily commercial, often prohibitively expensive for widespread academic use, and typically lack transparency in their underlying literature retrieval pipelines. SciArena is an open platform for automating scientific literature-grounded tasks based on a diverse collection of frontier foundation models. While its primary goal is to serve as an evaluation platform, our user study shows that it can also serve as a competitive alternative for assisting researchers with everyday literature tasks.

**Benchmarks for Scientific Literature-Grounded Tasks.** Extensive research has been dedicated to creating benchmarks for evaluating how well language models can address scientific literature understanding tasks. Historically, these benchmarks have often concentrated on specific, narrowly-defined tasks, such as evaluating performance on short-form question answering from document context [11, 75, 18, 35, 72, 36], generating summaries from multiple documents [45, 39, 33], as well as tasks like information extraction [41], hypothesis generation [40], and retrieval [16, 70]. Benchmarks such as SCHOLARQABENCH [8] were proposed to assess models on more complex, open-domain literature review tasks [2], offering a more realistic evaluation setting. While these benchmarks have provided valuable insights, they often fail to fully reflect the open-ended nature, diversity, and complexity of real-world research needs. which are difficult to anticipate and encode into static, curated benchmarks. Moreover, existing benchmarks primarily concentrate on a small number of well-resourced domains (*e.g.,* computer science and biomedicine), limiting the generalizability of evaluation and neglecting the needs of researchers in underrepresented fields. SciArena addresses these concerns by enabling real-time, open-ended, researcher-driven evaluation across a broad range of scientific domains, capturing diverse information needs that static benchmarks overlook.

**Foundation Model Evaluation via Human Preferences.** Traditional evaluation benchmarks often rely on automated metrics, which can be limiting in efficacy or capturing nuanced human judgments [9, 42, 84, 44]. Human evaluation is widely regarded as the gold standard for reliably evaluating foundation models [64, 84]. Thus, human preference-based evaluations have gained traction, particularly through the development of crowdsourcing platforms for pairwise model comparisons. One prominent example is Chatbot Arena [14], which set a precedent for collecting large-scale human votings to rank foundation models. Building on this approach, similar arena-based evaluation platforms have been introduced for multimodal foundation models [15, 46], generative models [32], text-to-speech models [54], search-augmented LLMs [51], and other complex tasks involving foundation models [74, 76, 82]. However, the scientific literature tasks are under-explored. Different from general-purpose tasks evaluated in existing arena platforms, scientific domain demand a high degree of domain expertise and precise literature retrieval for literature grounded generation. To address this gap, we carefully design SciArena platform and implement rigorous data quality controls to mitigate biases and address concerns raised by recent critiques of arena-style evaluations [71]. Furthermore, to advance the development of more reliable and human-aligned automated evaluation methods, we release SciArena-Eval, the first meta-evaluation benchmark for scientific literature tasks.

## 3   SciArena Platform

Our work introduces SciArena, an open platform for evaluating foundation models in their ability to understand, analyze, and synthesize scientific literature and generate citation-attributed answers to real-world research questions. In the following subsections, we describe the design of the SciArena

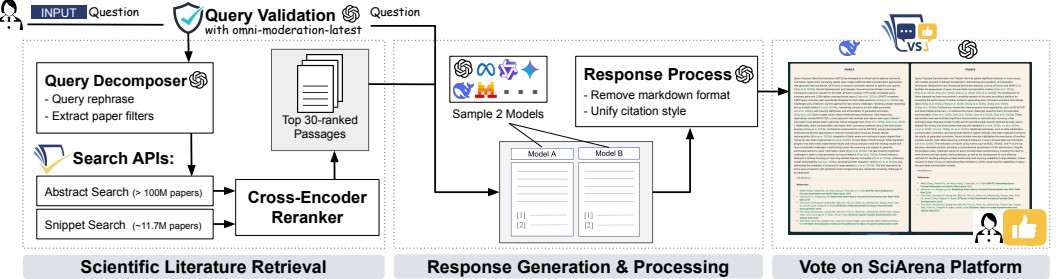

Figure 2: An overview of the SciArena interface pipeline.

platform in detail, including its response generation pipeline, Elo-based ranking system, and our strategies for addressing bias and the inherent challenges of human preference evaluation.

## 3.1 Platform Design

Figure 2 illustrates the overview of the SciArena interface pipeline. Upon receiving a user-submitted question, the SciArena interface first conducts a content moderation check to ensure the query is *not* potentially harmful.[2] If the query passes moderation, the interface uses the literature retrieval module from ScholarQA [69], a state-of-the-art agentic scientific literature synthesis system, to retrieve a set of relevant scientific paper contexts. These contexts, together with the original question, are then provided to two randomly selected foundation models drawn from a pool of strong open-source and proprietary frontier models. The two models independently generate responses grounded in the retrieved literature. Finally, the user is prompted to vote for their preferred response and may optionally provide a textual justification for their choice. We next detail the implementations of literature retrieval and model response generation.

**Scientific Literature Retrieval.** Unlike general-purpose tasks typically evaluated on other arena platforms, answering questions in the scientific domain requires responses grounded in scholarly literature. As such, the effectiveness of the literature retrieval pipeline is critical, as producing high-quality and informative responses depends on retrieving documents that are both topically relevant and of high scholarly credibility, which in turn sustains the usefulness of model outputs and encourages continued user participation with the SciArena platform. To support this goal, we apply the multi-stage retrieval system from ScholarQA [69], which integrates with the Semantic Scholar API and accesses a large, continuously updated scholarly corpus [3, 34]: over 100 million paper abstracts (abstract-level endpoint) and 11.7 million full-text papers (snippet-level endpoint[3] Specifically, given a user query, a strong LLM (GPT-4o prior to April 20; GPT-4.1 afterward) generates queries tailored to each endpoint and extracts any user-defined metadata filters (*e.g.,* publication years, authors, venues). Following a similar setup to ScholarQA, the system retrieves up to 40 paper snippets from full-text and 20 abstracts, which are then reranked using a state-of-the-art re-ranker [66]. The top-30 results are selected as contextual input for the two randomly sampled foundation models.

**Model Response Generation and Postprocessing.** Given the user question and 30 relevant paper contexts obtained from the literature retrieval module, the foundation model is prompted to generate a citation-attributed response that well addressed the user needs. This step typically demands multi-document reasoning and synthesizing information across all the retrieved sources. The model also should decide which sources of information should include and cite in the final generated response. Recent studies have shown that stylistic elements in the response—such as markdown formatting (*e.g.,* bold text, bullet points) or the use of emojis—can inadvertently influence user preferences during evaluation [14, 23, 15]. While such formats can be favored in general-purpose applications, they are uncommon in scientific literature settings. In practice, we observe that some evaluated foundation models tend to adopt these styles. To mitigate potential biases introduced by response formatting, we explicitly prompt the model to generate plain-text responses without markdown. Additionally, we use a strong LLM (GPT-4o prior to April 20; GPT-4.1 afterward) in a postprocessing step to ensure

---

[2]Content moderation is performed using OpenAI's `omni-moderation-latest` model.
[3]Paper snippets are passages (up to about 480 tokens) that the Semantic Scholar snippet-search endpoint automatically extracts from a paper's abstract or main body. Multiple snippets are indexed for every paper.

formatting consistency. The prompts used for model response generation and response postprocessing are provided in Appendix B.3.

**Evaluated Foundation Models.** As of the evaluation cutoff date, Jan 11, 2026, SciArena hosts a total of 47 frontier foundation models for evaluation. This set includes 25 proprietary models and 22 open-source models chosen for their strong representation of current state-of-the-art capabilities. The collection features reasoning models such as o4-mini [60], DeepSeek-R1 [22], and QwQ-32B [63], alongside hybrid reasoning models like Claude-4.1-series [6] and Qwen3-series [62] models. For reasoning models, we remove the thought process content preceding the thinking tokens during postprocessing; for hybrid reasoning models, we enable their "thinking" mode. We detail the model configurations in Appendix B.4. We will be continuously adding new models to the SciArena platform to ensure ongoing evaluation of the latest advancements.

## 3.2 User Study on SciArena vs. Commercial Platforms for Scientific Literature Tasks

An arena platform relies on user contributions, and therefore must deliver clear value and meet fundamental usability standards. To investigate this, we conducted a user study involving four researchers across different scientific domains, comparing SciArena with leading commercial platforms for performing routine scientific literature tasks in the context of their real-world research workflows. Specifically, we evaluated two major categories of alternatives: (1) *Chatbot Platforms* with search capabilities, including ChatGPT with Search and Perplexity AI; and (2) *Agent-based Platforms* designed for complex, multi-step research tasks, including OpenAI Deep Research and Gemini Deep Research. All platforms were tested using their latest publicly available versions as of May 5, 2025.

Each participant spent a total of 60 minutes using each platform and provided feedback. Participant biographies and detailed responses are available in Appendix B.1. In summary, compared to ChatGPT and Perplexity with search capabilities, participants found that the cited papers in SciArena are more relevant, while other platforms occasionally cite less reliable sources such as blogs or media articles, which participants considered less trustworthy. When compared to the two Deep Research platforms, SciArena is more efficient, with shorter wait times. It achieves comparable performance on well-defined questions; while for exploratory queries, the longer reports generated by Deep Research platforms sometimes contains more useful information. The study results indicate that SciArena is an effective standalone tool for high-quality literature analysis, with all participants expressing interest in continued use of SciArena.

## 3.3 Leaderboard Ranking with Elo Rating

Following Chatbot Arena, we adopt the Bradley–Terry (BT) model [10] for Elo rating estimation [26].

**Bradley–Terry (BT) Model.** Unlike the standard online Elo rating system (described in Appendix B.2), which can be sensitive to the order of comparisons, the BT model provides a more robust way to estimate model strengths by fitting a logistic regression to the outcomes of all pairwise comparisons. Let $n$ denote the total number of pairwise comparisons and $M$ the number of models. For each comparison $i \in [n]$, we define $X_i \in \mathbb{R}^M$ as a feature vector where $X_{i,m} = 1$ if model $m$ appears first, $X_{i,m} = -1$ if it appears second, and 0 otherwise; and $Y_i \in \{0, 1\}$ as the outcome, where 1 indicates that the first model wins. It estimates a strength vector $\beta \in \mathbb{R}^M$ by minimizing the average cross-entropy loss:

$$\hat{\beta} = \arg \min_{\beta \in \mathbb{R}^M} \frac{1}{n} \sum_{i=1}^{n} \text{CE}(\sigma(X_i^\top \beta), Y_i), \tag{1}$$

where $\sigma$ is the sigmoid function and CE denotes the cross-entropy loss. The resulting coefficients $\hat{\beta}$ serve as the estimated Elo ratings, which determine the final leaderboard rankings of the models. Since this modeling does not consider ties, in practice, we duplicate all the votes and force half of the tie votes to be counted as first model winning ($Y_i = 1$) and the other half as second model winning ($Y_i = 0$). To further investigate the variance of the estimated Elo rating, we apply bootstrapping with 100 resamples to compute confidence intervals for each rating.

**Controlling Stylistic Biases in Evaluation.** Recent studies have highlighted potential confounding factors in model evaluation, such as response length and stylistic formatting [14, 44, 24]. To assess

| Voting |
| --- |
| Total Votes: 20,832 |
| "A" : "B" : "Tie" : "Both Bad" = 9,107 : 9,392 : 1,746 : 587 |

**Question Category with Examples**

*Conceptual Explanation* (32.94%): How do locally described freeform surfaces impact optical system design flexibility?

*State-of-the-Art Assessment* (25.36%): What new trends emerge in digital philology across various textual traditions?

*Challenges & Limitations* (23.20%): What are the current challenges in characterizing large RNA molecules for therapeutic development?

*Methodology Inquiry* (8.76%): How can the integration of different renewable energy sources be achieved to ensure a stable power supply?

*Paper Finding* (4.78%): Please find papers on solving the project scheduling problems using reinforcement learning.

*Others* (4.96%)

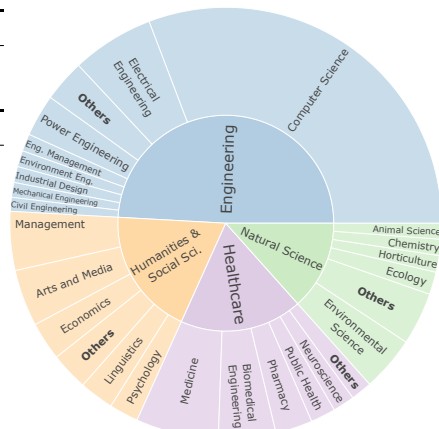

Figure 3: Statistics of the initial human preference data collected through SciArena, including voting information and distribution across question categories and scientific disciplines.

the influence of these factors in SciArena evaluations, we follow prior work that extends the BT model to incorporate style features [37] and present our findings in Section 5.2. Specifically, given a set of style features (*e.g.,* model response length), we augment the BT model with a style vector $\vec{Z}$, where each $Z_i \in \mathbb{R}^S$ represents the $S$-dimensional style feature vector for instance $i$. The extended model has the style coefficients $\gamma \in \mathbb{R}^S$:

$$\hat{\beta}, \hat{\gamma} = \arg \min_{\beta \in \mathbb{R}^M, \gamma \in \mathbb{R}^S} \frac{1}{n} \sum_{i=1}^n \mathrm{CE}(\sigma(X_i^\top \beta + Z_i^\top \gamma), Y_i)$$

The resulting $\hat{\gamma}$ quantifies the influence of style features on user preferences.

## 4 SciArena Data

This section describes our data collection details and quality analysis of the collected data.

### 4.1 SciArena Human Preference Data Collection

Existing arena-based evaluation platforms [14, 32, 46, 74, 76, 82] typically gather human preference data from a broad, non-expert user base. While effective for general-purpose tasks, these methods are inadequate for our focus on scientific literature tasks, which require domain expertise. To address this, we carefully design our data collection process to ensure the high quality of collected human preference data thus maintaining a reliable SciArena leaderboard.

**Initial Data Collection.** Our initial data collection phase (beta release of SciArena) involves 102 researchers working across four core disciplines: Natural Science, Healthcare, Humanities & Social Sciences, and Engineering. Anonymized profiles of the participating researchers are presented in Appendix C.1. Each expert researcher involved has authored at least two peer-reviewed publications and has prior experience with AI-assisted literature tools. Before beginning the annotation process, all annotators complete a comprehensive, one-hour training session conducted by a member of our team to ensure consistency and accuracy across evaluation. We collect a total of 20,832 votes.

**Data Collection From Scientific Community.** While SciArena is publicly accessible, only votes from users who (1) pass anomaly detection checks and (2) consent to our terms of use, including data collection, are included in the final leaderboard. We follow the same protocol as Chatbot Arena [14] for identifying anomalous users, as detailed in Appendix B.5.

### 4.2 Data Analysis

Figure 3 presents the statistics of the voting data collected through SciArena. We collect a total of 20,832 votes. To better understand the types of questions researchers pose on the platform, we

randomly sample 200 questions for manual analysis and organize them into five primary categories, along with an "Others" category, as shown in Figure 3. We then use GPT-4.1 to classify all collected questions (using the prompt described in Appendix C.3). The resulting distribution of question categories, along with corresponding examples, is shown in Figure 3. Overall, users primarily use SciArena to gain a deeper understanding of scientific concepts and to explore the current state of progress, ongoing challenges, and open questions within a scientific domain.

### 4.3 Quality Assessment of Collected Human Votes

To evaluate the reliability of our collected preference data, we conducted a quality assessment of votes obtained from both the initial data collection stage and the scientific community.

**Quality Assessment of Internal-Annotated Votes.** For voting data collected in the *initial data collection* stage, we assess two metrics: (1) inter-annotator agreement (IAA) and (2) self-consistency, which reflects the internal stability of individual annotators' judgments over time. We use accuracy and weighted Cohen's $\kappa$ as metrics. For each scientific discipline, we randomly selected 100 questions and the corresponding pair of two model responses, resulting in a total of 400 examples.

To measure IAA, each example was independently evaluated by a second expert with a closely aligned research background. To measure self-consistency, annotators re-evaluated examples they had previously annotated after a minimum interval of two weeks. As shown in Table 3, the self-consistency results demonstrate that expert preferences remain stable over time, indicating decisions are not influenced by momentary bias. Similarly, the high IAA shows that despite the subjective nature of some questions, experts still tend to reach similar judgments. These results confirm the high quality and reliability of the collected votes.

Table 1: Inter-annotator agreement (IAA) and self-consistency (SC) across disciplines. We report accuracy and weighted Cohen's $\kappa$.

| Discipline | IAA | | SC | |
|---|---|---|---|---|
| | Acc | $\kappa$ | Acc | $\kappa$ |
| Natural Science | 0.82 | 0.76 | 0.94 | 0.91 |
| Healthcare | 0.87 | 0.82 | 0.91 | 0.89 |
| Humanity & Social Sci. | 0.78 | 0.70 | 0.96 | 0.94 |
| Engineering | 0.82 | 0.75 | 0.93 | 0.91 |
| Average | 0.82 | 0.76 | 0.94 | 0.91 |

**Quality Assessment of Community-Contributed Votes.** To validate the quality of the collected votes from the research community, we randomly sampled 150 examples from the newly collected data and assigned 50 examples to each of three authors for manual review. For 17 of these, the reviewers were unable to make a relevance judgment due to unfamiliarity with the content. For the remaining 133 examples, each was labeled according to the following categories: (1) Clearly reasonable vote: The vote is clear and would be widely agreed upon. (2) Vague vote: The selected vote is acceptable, but alternative votes could also be justified. (3) Wrong vote: The voting outcome is likely to be disagreed with by most reviewers. We revealed that 73.7% (98 / 133) of votes are clearly reasonable, 17.3% (23 / 133) are acceptable but somewhat ambiguous, and 9.0% (12 / 133) were judged as incorrect. These results suggest that the community-contributed votes are also reliable.

## 5 SciArena Leaderboard Analysis

Table 2 presents the SciArena leaderboard across scientific disciplines. A detailed breakdown of Elo ratings by question category and Elo rating confidence intervals are provided in Appendix D.1.

### 5.1 Main Results

o3, Claude-4.1-Opus, and GPT-5 emerge as the top-3 models overall. Model performance varies notably across domains: for instance, o3 attains the highest score in *Engineering*, while GPT-5 leads in the disciplines of *Natural Science* and *Humanities & Social Science*. A similar pattern emerges in fine-grained analyses by question type (Appendix D.1): Claude-4.1-Opus excels at questions concerning challenges and limitations, GPT-5 shows a clear advantage on paper findings, and the o3 models demonstrate consistently strong performance across all categories. Among open-source models, GPT-OSS-120B and Deepseek-R1-0528 stand out, ranking 8th and 9th overall, respectively, and outperforming several strong proprietary models such as o4-mini and Claude-4-Sonnet.

Table 2: SciArena Leaderboard. Elo scores of evaluated models, both overall and within four scientific disciplines, as of the evaluation cutoff date at Jan 11, 2026. Only models with at least 100 votes are included. The top and second-best models are marked in **bold** and underlined, respectively. The highest-ranking proprietary and open-source models are color-highlighted.

| Models | Release | Battles | Scientific Discipline | | | | Cost per 100 Calls (USD) | Elo Score |
| | | | Natural Science | Healthcare | Humanities & Social Sci. | Engineering | | |
|---|---|---|---|---|---|---|---|---|
| o3 [60] | 2025-04 | 1076 | **1185.1** | 1153.6 | 1117.9 | **1176.7** | 3.54 | **1151.4** |
| Claude-4.1-Opus [5] | 2025-07 | 341 | 1155.3 | 1071.7 | **1173.3** | 1138.0 | 28.65 | 1126.4 |
| GPT-5 [58] | 2025-06 | 357 | 1110.5 | 1121.7 | 1112.4 | 1133.5 | 2.98 | 1122.5 |
| Gemini-3-Pro-Preview [30] | 2025-08 | 215 | 1063.8 | **1162.2** | 1111.4 | 1082.4 | 4.25 | 1086.3 |
| GPT-5.1 [56] | 2025-08 | 200 | 1161.0 | 1054.8 | 995.8 | 1077.4 | 2.98 | 1079.7 |
| Claude-4-Opus [6] | 2025-05 | 1403 | 1000.3 | 1109.3 | 1070.7 | 1084.2 | 28.45 | 1071.7 |
| GPT-5-mini [58] | 2025-07 | 342 | 1125.3 | 1093.1 | 1104.6 | 1073.9 | 0.59 | 1067.1 |
| Gemini-2.5-Pro [29] | 2025-06 | 1253 | 1080.9 | 1103.4 | 996.8 | 1086.0 | 2.87 | 1061.3 |
| Grok-4 [80] | 2025-07 | 465 | 1016.5 | 1017.4 | 1044.8 | 1056.9 | 5.73 | 1044.6 |
| Deepseek-R1-0528 [21] | 2025-05 | 1340 | 1035.9 | 1045.1 | 1031.6 | 1058.8 | 0.79 | 1041.8 |
| GPT-OSS-120B [59] | 2025-08 | 350 | 1011.0 | 1093.6 | 1007.3 | 1041.7 | 0.10 | 1036.5 |
| Qwen3-235B-A22B-Thinking-2507 [62] | 2025-07 | 381 | 1116.4 | 1014.4 | 1004.4 | 1030.5 | 0.24 | 1036.3 |
| o4-mini [60] | 2025-04 | 1712 | 971.5 | 1063.2 | 981.3 | 1056.0 | 2.48 | 1028.6 |
| Claude-4-Sonnet [6] | 2025-05 | 1336 | 1030.0 | 1024.0 | 1021.9 | 1030.9 | 5.70 | 1025.2 |
| Qwen3-235B-A22B-2507 [62] | 2025-07 | 263 | 1006.1 | 1035.6 | 1090.9 | 1023.5 | 0.21 | 1020.5 |
| GPT-4.1 [57] | 2025-04 | 2011 | 960.2 | 1009.4 | 1030.7 | 1031.4 | 2.73 | 1018.6 |
| GPT-4.1-mini [57] | 2025-04 | 1416 | 999.4 | 1024.0 | 967.8 | 1031.9 | 0.55 | 1014.6 |
| Qwen3-30B-A3B-Instruct-2507 [62] | 2025-07 | 375 | 994.1 | 936.3 | 945.7 | 1007.8 | 0.28 | 1010.2 |
| Gemini-2.5-Pro-Preview [28] | 2025-03 | 1044 | 965.9 | 1059.1 | 988.1 | 975.4 | 2.93 | 1009.7 |
| GLM-4.5 [90] | 2025-06 | 371 | 1011.4 | 999.6 | 968.2 | 1025.5 | 1.00 | 1006.9 |
| Deepseek-R1 [22] | 2025-01 | 1814 | 1020.7 | 1010.7 | 976.7 | 1032.1 | 0.74 | 1006.9 |
| Deepseek-V3 [20] | 2025-03 | 1989 | 1010.5 | 1018.3 | 1013.9 | 988.2 | 0.37 | 1003.5 |
| Qwen3-235B-A22B [62] | 2025-04 | 1672 | 1002.1 | 1000.0 | 969.8 | 995.8 | 0.36 | 1002.6 |
| Kimi-K2 [53] | 2025-07 | 501 | 934.7 | 1045.4 | 989.1 | 1007.8 | 0.75 | 1001.9 |
| Grok-3 [79] | 2025-02 | 1935 | 935.0 | 980.7 | 1000.4 | 993.3 | 4.14 | 989.3 |
| QwQ-32B [63] | 2025-03 | 1775 | 980.0 | 987.9 | 952.0 | 979.7 | 0.11 | 977.5 |
| Claude-3-7-Sonnet [4] | 2025-02 | 1961 | 1006.4 | 930.3 | 966.1 | 973.0 | 5.74 | 966.4 |
| Gemini-2.5-Flash [31] | 2025-05 | 1472 | 982.8 | 940.0 | 972.0 | 966.4 | 0.71 | 966.0 |
| Olmo-3.1-32B-Instruct [1] | 2025-08 | 128 | 844.2 | 963.4 | 927.5 | 1039.2 | 0.17 | 963.6 |
| Qwen3-32B [62] | 2025-04 | 1662 | 968.8 | 942.8 | 953.7 | 936.8 | 0.17 | 962.7 |
| Gemini-2.5-Flash-Preview [31] | 2025-04 | 1370 | 885.6 | 904.6 | 978.7 | 917.2 | 0.72 | 932.8 |
| GPT-OSS-20B [59] | 2025-08 | 370 | 924.5 | 915.9 | 1000.7 | 919.1 | 0.05 | 927.6 |
| GPT-5-nano [58] | 2025-07 | 330 | 965.3 | 926.4 | 898.5 | 892.2 | 0.12 | 902.6 |
| Mistral-Small-3.1 [52] | 2025-03 | 1663 | 915.8 | 915.1 | 892.8 | 881.4 | 0.06 | 889.8 |
| Mistral-Medium-3 [50] | 2025-05 | 1761 | 874.8 | 890.8 | 837.6 | 888.5 | 0.65 | 884.8 |
| Minimax-M1 [49] | 2025-06 | 726 | 917.6 | 958.0 | 869.0 | 858.6 | 0.57 | 879.9 |
| Llama-4-Maverick [48] | 2025-04 | 1749 | 831.3 | 872.8 | 888.2 | 801.4 | 0.20 | 844.3 |
| Llama-4-Scout [48] | 2025-04 | 2112 | 873.8 | 815.3 | 847.6 | 791.3 | 0.11 | 829.8 |

## 5.2 Preference Analyses in SciArena Evaluation

Citations play a central role in scientific literature tasks and are a distinctive feature of SciArena. Contemporary with our work, Search Arena [51] finds that (1) users tend to prefer responses containing more references, and (2) that their judgments can be swayed by the mere presence of citations, regardless of whether those citations are properly attributed to specific claims. In this subsection, we examine how citation-related features influence user preferences in SciArena, focusing on two key dimensions: number of citations and attribution of inline citations to the generated content. We also analyze the feature of response length in SciArena evaluation. The implementation details of preference analyses are provided in Appendix D.2.

**Citation Count.** We investigate the influence of citation count on user preferences within the SciArena framework. Our analysis yields a modest but positive Bradley-Terry coefficient for citation count ($\gamma = 0.039$). In comparison, Search Arena [51] reports a substantially higher coefficient ($\gamma = 0.209$), indicating a stronger user bias toward citation-rich outputs. These findings suggest that citation count is not a dominant factor in shaping preferences in the SciArena evaluation setting.

**Citation Attribution.** We further examine how the correctness of citation-to-claim attribution impacts user preferences. Using the o4-mini model, we classify each citation-response pair into two categories: *supporting*, where the citation backs the response, and *irrelevant or contradicting*, where the citation does not substantiate the claim or directly contradicts it. We observe a statistically positive coefficient for supporting citations ($\gamma = 0.155$) and a negative coefficient for irrelevant or

contradicting ones ($\gamma = -0.154$). These findings underscore a difference between SciArena and general-purpose retrieval settings. Specifically, in Search Arena, users tend to prefer responses with more citations regardless of attribution quality[4]. In contrast, user behavior in SciArena indicates a clear preference for citations that are highly relevant and correctly attributed to the response content.

**Response Length.** In Section 3.1, we describe our efforts to postprocess model outputs into a standardized, literature-based format with citations, aiming to reduce bias introduced by model-specific response styles. However, response length remains a potential confounding factor in evaluations. To explore this further, and following prior work [14, 23], we analyze the effect of response length features in SciArena, as described in Section 3.3. The resulting Bradley-Terry coefficient for response length is $\gamma = 0.141$, which is substantially lower than those reported in Chatbot Arena ($\gamma = 0.25$) [14], Vision Arena ($\gamma = 0.27$) [15], and Search Arena ($\gamma = 0.33$) [51]. These findings suggest that SciArena exhibits reduced length bias and yields more reliable human preference data.

### 5.3 Case Analysis

**Analysis of o3 Model.** To better understand the strengths of current best-performing models, we conducted a human evaluation by sampling 200 voting examples that compare o3 with other top-performing models (*e.g.,* Claude-4-Opus and Gemini-2.5-Pro), with a focus on Engineering. Our analysis highlights four key strengths of the o3 model: (1) **more detailed elaboration on cited papers**: the o3 model consistently provides deeper explanations and richer technical insights drawn from referenced literature; (2) **more professional and precise terminology**: the o3 model tends to employ domain-specific vocabulary and technically accurate phrasing, reducing ambiguity and enhancing clarity; (3) **clear structured presentation**: o3's responses are better organized, improving both readability and synthesis of complex information; and (4) **more comprehensive coverage**: for question types like *Challenges & Limitations* and *State-of-the-Art Assessment*, o3's responses are notably more comprehensive, addressing a broader range of points likely to be of interest to users. Examples and analyses for each strength are provided in Appendix D.3.

**Analysis of Model Failure Case.** We then use the collected data to analyze examples that are especially challenging for current foundation models. Specifically, we filter for instances where (i) users judged both two models' responses to be poor, because such examples expose limitations shared across multiple systems and reveal systematic weaknesses; or (ii) the two responses are from one of the top-3 models and one of the other models, and the response from a top-3 model was voted as worse, because these cases help us understand the limitations of the best-performing models. From this filtered set, we sample 100 examples for detailed human analysis. We categorize the most common failure modes into five types: (1) **failure to answer the question**, where responses skirt main points, offering tangential or irrelevant information instead; (2) **conflict with cited papers**, where references are misinterpreted or misaligned, producing claims unsupported by original studies; (3) **lack of detail**, where answers mention headlines only, omitting necessary mechanisms, context, examples, and quantification; (4) **misunderstanding of terminology**, where key terms are redefined or confused, leading to misaligned metrics and flawed conclusions; and (5) **incoherent structure**, where content lacks logical flow, mixing unrelated points without transitions, hindering comprehension and synthesis. Examples and detailed analyses for each failure type are provided in Appendix D.4.

## 6 SciArena-Eval for Evaluating Model-based Evaluators

### 6.1 SciArena-Eval Benchmark

As discussed in the Related Work section, developing model-based evaluation methods for literature-based long-form answers remains a significant challenge. This issue is further exacerbated by the absence of a standardized meta-evaluation benchmark for comparing different evaluation approaches.

To address this gap, we introduce the SciArena-Eval benchmark, constructed using human preference data collected via SciArena. Specifically, we randomly sample 500 voting instances (250 examples where annotators preferred "model A" and 250 where they preferred "model B") per scientific discipline, resulting in a total of 2,000 examples in SciArena-Eval. Votes marked as ties are excluded

---

[4]The Search Arena paper reports $\gamma_{\text{support}} = 0.29$, $\gamma_{\text{irrelevant}} = 0.27$, suggesting users do not differentiate between supporting and irrelevant citations.

from the benchmark, as they do not provide a definitive signal about which model is better, making them unsuitable for evaluating discriminative ability. The automated evaluation system is tasked with identifying the superior model response for a given literature-based questions. We assess system performance by measuring accuracy against expert human judgments.

## 6.2 Experiments on SciArena-Eval

We assess model-based pairwise evaluation protocols, where an evaluator model is given a question and two candidate responses and must select the better one (with prompt provided in Appendix E.1). Our evaluation covers a range of proprietary frontier models as well as open-source models.

As shown in Table 3, **SciArena-Eval presents significant challenges for model-based evaluators.** Even the best-performing model, o3, achieves only 65.1% accuracy. Lower-performing models, such as Gemini-2.5-Flash-Preview and Llama-4-sereis models, perform only slightly better than random guessing. Notably, similar pairwise evaluation protocols have shown strong alignment with human judgments (*i.e.,* exceeding 70% correlation) on general-purpose benchmarks like AlpacaEval [38] and WildChat [86]. The comparatively low accuracy on SciArena-Eval highlights the unique difficulty of evaluating scientific reasoning tasks. **Reasoning-augmented models generally outperform their non-reasoning counterparts from the same organization.** For instance, o4-mini surpasses GPT-4.1 by 2.9%, and DeepSeek-R1 outperforms DeepSeek-V3 by 0.7%. These results highlight the effectiveness of inference-time scaling in improving evaluation performance. We believe that SciArena-Eval can serve as a robust benchmark for the development and evaluation of automated systems for scientific literature tasks in future research.

Table 3: SciArena-Eval Results.

| Base Model | Acc |
|---|---|
| Random Guess | 50.0 |
| o3 | **65.1** |
| o4-mini | 64.8 |
| GPT-5 (*high* reasoning) | 63.2 |
| GPT-4.1 | 61.9 |
| DeepSeek-R1 | 61.2 |
| GPT-5 (*medium* reasoning) | 60.6 |
| DeepSeek-V3 | 60.5 |
| GPT-4.1-mini | 60.5 |
| Gemini-2.5-Pro-Preview | 60.3 |
| Claude-3.7-Sonnet | 59.2 |
| Qwen3-32B | 58.1 |
| Gemini-2.5-Flash-Preview | 57.8 |
| Llama-4-Scout | 57.7 |
| Llama-4-Maverick | 57.5 |

## 7 Discussion

We introduce SciArena, a dynamic evaluation platform designed to compare foundation models on non-verifiable scientific literature-grounded tasks. By collecting over 20,000 votes from human researchers, we provide a rich dataset for analyzing human preferences across models. Our comprehensive analysis reveals key insights and highlights promising directions for advancing foundation models in scientific literature-grounded tasks. Moreover, we propose the SciArena-Eval benchmark to evaluate the model-based automated evaluation systems. Our experiments highlight the benchmark's challenges and emphasize the need for more robust and reliable automated evaluation methods.

**Open-source Efforts.** SciArena is freely accessible to the public. We have released all collected human preference data, offering a valuable resource for understanding human judgment in scientific literature tasks. This data supports the development of models that better align with human standards in real-world research scenarios. The newly curated SciArena-Eval benchmark is also publicly available. Additionally, by open-sourcing the SciArena code, we enable researchers and developers to adapt our methods (*e.g.,* literature retrieval pipeline, arena platform) to other tasks.

**Mitigating Challenges and Biases in SciArena Evaluation.** We design SciArena and data collection pipeline with bias mitigation as a primary objective. SciArena includes only models with publicly accessible APIs or checkpoints, thereby eliminating advantages stemming from private, unreleased models. As explained in Section 4.1, only votes from users passing anomalous check are included in the final leaderboard, alleviating concerns about the reliability of open crowdsourcing. In Section 4.3, we present analyses of inter-annotator agreement and annotator self-consistency, both of which demonstrate strong alignment in user preferences. Additionally, our analysis in Section 5.2 shows that the data collected in SciArena reflects a clear user preference for citations that are relevant and accurately attributed to the response content, with minimal influence from citation count.

## Limitations and Future Work

While our platform integrates a wide range of state-of-the-art foundation models for convenient comparison, it currently excludes some older or deprecated versions, such as GPT-4o, Llama-3.1, and Qwen-2.5. This omission may pose challenges for developers seeking to benchmark the latest models against their earlier counterparts. Moving forward, we plan to incorporate newly released models into SciArena on a rolling basis to ensure broader and more consistent coverage. Additionally, SciArena is designed to be compatible with agent-based literature review tools such as OpenAI's Deep Research and Gemini's Deep Research. However, due to limitations such as daily usage caps and the unavailability of service APIs, we are currently unable to integrate them into our platform.

## Acknowledgments

This project is supported in part by Defense Advanced Research Projects Agency's (DARPA) SciFy program (Agreement No. HR00112520300), and Google's "Scholar Research Awards" program. We acknowledge the National Artificial Intelligence Research Resource (NAIRR) Pilot and Microsoft Azure for contributing to the results in this work.

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

## Appendix Contents


# B    SciArena Platform

## B.1    User Study Results

Table 4: User feedback comparing the usability of SciArena with that of other platforms for conducting scientific-literature tasks

| **Question 1:** Compare SciArena and Chatbot Platforms with search ability like ChatGPT *search* and Perplexity AI in the aspect of usability of output and generation time. | **Question 2:** Compare SciArena and Agent-based Platforms like ChatGPT *deep research* and Gemini *deep research* in the aspect of usability of output and generation time. |
|---|---|
| **User 1 [clinical researcher in public health system]:**
ChatGPT search and Perplexity AI can give good answers for some simple questions. But for complex queries (e.g. treatment comparisons or emerging technologies) SciArena captures my intention better and provides a more structured, comprehensive answer. And both of them could provide a fast response | **User 1 [clinical researcher in public health system]:**
ChatGPT deep research and Gemini deep research are great for a well-organized overview. When preparing materials for publication, SciArena helps more with traceable citations and compact answers. And SciArena usually gives me a more prompt response. |
| **User 2 [4th-year PhD Student in Computer Science]:**
I find SciArena valuable for paper writing with fewer hallucinations. Conversely, Chatbot Platforms with search ability sometimes gives unsupported statements citing unreliable resources. SciArena sometimes takes a little bit longer to generate the response. But, I am fine with it since it's from reliable resources. | **User 2 [4th-year PhD Student in Computer Science Engineering]:**
Both tools give grounded, comprehensive material, and I find them equally helpful for my study. Regarding on the generation time, SciArena is more efficient in Literature Review. |
| **User 3 [3rd-year PhD Student in Environmental Science]:**
Chatbot Platforms with search ability is great for general queries, but for niche topics it can surface broad or outdated sources. SciArena feels more trustworthy for academic work. And both of them are fast considering my questions. | **User 3 [3rd-year PhD Student in Environmental Science]:**
Agent-based Platforms provides a longer, narrative-style answer, but can be too lengthy for simple queries. And It usually takes several minutes to get a comprehensive report. SciArena offers more curated answers so I can quickly decide which papers are worth reading. |
| **User 4 [Applied Economist in Humanity]:**
ChatGPT search like tool can be quick but surface-level. SciArena gives claims tied directly to the question, with proper citations. Both provide valuable information overall. And both provide answers within a short of time. | **User 4 [Applied Economist in Humanity]:**
ChatGPT Deep Research like tool sometimes over-explains. When I just need 2–3 key papers, SciArena is leaner and helps me stay focused. And ChatGPT Deep Research like tool often takes a while to generate those report-like responses. |

Table 5: Likelihood of continued SciArena platform use (Q3)

| **Question 3:** From 1 to 10, how likely are you to continue using this platform in your daily research? (Optional reasons below) |
|---|
| **User 1 [clinical researcher in public health system]: 9/10**
**User 2 [4th-year PhD Student in Computer Science Engineering]: 8/10**. Relies on ChatGPT Deep Research for speculative reasoning and ChatGPT search for broader overviews.
**User 3 [Natural Science]: 8/10**. SciArena makes some projects easier, especially when hard-science support is needed.
**User 4 [Humanity]: 7/10**. It's fast, reliable, and doesn't try to be overly smart. |

## B.2 Online Elo Rating System

The online Elo rating system models the probability that model $i$ wins against model $j$ given their current ratings $R_i$ and $R_j$, where $i, j \in N$. Specifically, we define a binary outcome $Y_{ij}$, where $Y_{ij} = 1$ if model $i$ wins and $Y_{ij} = 0$ otherwise. The win probability is modeled as:

$$P(Y_{ij} = 1) = \frac{1}{1 + 10^{(R_j - R_i)/\alpha}}, \tag{2}$$

where $\alpha = 400$ is the scaling factor. After each match, player ratings are updated as follows:

$$R_i' = R_i + K \cdot (S(i|j) - E(i|j)), \tag{3}$$

where $S(i|j)$ is the actual match result (1 for win, 0.5 for tie, 0 for loss), $K$ determines the sensitivity of the Elo system to new match results, and $E(i|j) = P(Y_{ij} = 1)$ is the expected score. This update rule ensures that higher-rated models gain fewer points when winning and lose more points when losing, while lower-rated models behave oppositely. However, the online Elo ratings are sensitive to the order in which comparisons occur.

## B.3 Prompts for Model Response Generation and Postprocessing

---

**[System Prompt]**
You are a helpful research assistant tasked with producing a citation-attributed response to a user's question. Please follow these instructions:

1. Inputs
- References: A list of papers (each with a title and a brief context). Note that some papers may be irrelevant to the user question.
- User question: A question or topic the user wants to investigate.

2. Your Task
- Examine all provided References and select the most relevant papers that directly address the user question.
- Write a citation-attributed response that addresses the user question.

3. Citation Format
- Cite each paper using square brackets with the paper's index from the References list (e.g., "Several studies suggest X [1].").
- Do not cite papers not listed in the References.

**[User Prompt]**
References List (list of paper that might be relevant):

1. Title: {paper-1-title}
Authors: {paper-1-authors}
Relevant Context: {paper-1-context}

2. Title: {paper-2-title}
Authors: {paper-2-authors}
Relevant Context: {paper-2-context}

(...abbreviated...)

Question: {question}

Write a citation-attributed response that addresses the user question.

---

Figure 4: The prompt used for model response generation.

Figure 5: The prompt used for model response postprocessing using the GPT-4.1 model. Citations are then matched to the reference list using rule-based methods, and the finalized response is displayed in the SciArena interface.

## B.4 Configurations of Evaluated Foundation Models, As of Jan 11, 2026

Table 6: Details of the foundation models included in SciArena as of the date at Jan 11, 2026.

| Organization | Model | Release | Version | Reasoning Model? |
|---|---|---|---|---|
| *Proprietary Models* | | | | |
| OpenAI | o3 | 2025-04 | `o3-2025-04-16` | yes |
| | o4-mini | 2025-04 | `o4-mini-2025-04-16` | yes |
| | GPT-4.1 | 2025-04 | `gpt-4.1-2025-04-14` | no |
| | GPT-4.1-mini | 2025-04 | `gpt-4.1-mini-2025-04-14` | no |
| | GPT-5 | 2025-08 | `gpt-5-2025-08-07` | yes |
| | GPT-5 Mini | 2025-08 | `gpt-5-mini-2025-08-07` | yes |
| | GPT-5 Nano | 2025-08 | `gpt-5-nano-2025-08-07` | yes |
| | GPT-5.1 | 2025-11 | `gpt-5.1-2025-11-13` | yes |
| Google | Gemini 3 Pro Preview | 2025-11 | `gemini-3-pro-preview` | hybrid |
| | Gemini 2.5 Pro | 2025-06 | `gemini-2.5-pro` | hybrid |
| | Gemini 2.5 Flash | 2025-06 | `gemini-2.5-flash` | hybrid |
| | Gemini 2.5 Pro Preview | 2025-03 | `gemini-2.5-pro-preview-03-25` | hybrid |
| | Gemini 2.5 Flash Preview | 2025-04 | `gemini-2.5-flash-preview-04-17` | hybrid |
| Anthropic | Claude 4.1 Opus | 2025-08 | `claude-opus-4-1-20250805` | hybrid |
| | Claude 4 Opus | 2025-05 | `claude-opus-4-20250514` | hybrid |
| | Claude 4 Sonnet | 2025-05 | `claude-sonnet-4-20250514` | hybrid |
| | Claude 4.5 Sonnet | 2025-09 | `claude-sonnet-4-5-20250929` | hybrid |
| | Claude 3.7 Sonnet | 2025-02 | `claude-3-7-sonnet-20250219` | hybrid |
| xAI | Grok3 (Beta) | 2025-02 | `grok-3` | no |
| | Grok 4 | 2025-07 | `grok-4-0709` | hybrid |
| | Grok 4 Fast Reasoning | 2025-09 | `grok-4-fast-reasoning` | yes |
| | Grok 4 Fast Non-Reasoning | 2025-09 | `grok-4-fast-non-reasoning` | no |
| Mistral AI | Mistral-Medium-3 | 2025-05 | `Mistral-Medium-3-45B-Instruct-2505` | no |
| MoonshotAI | Kimi K2 | 2025-07 | `kimi-k2` | no |
| | Kimi K2 (0905) | 2025-09 | `kimi-k2-0905` | no |
| *Open-source Multimodal Foundation Models* | | | | |
| Mistral AI | Mistral-Small-3.1 | 2025-03 | `mistralai/Mistral-Small-3.1-24B-Instruct-2503` | no |
| Alibaba | Qwen3-235B-A22B | 2025-04 | `Qwen3-235B-A22B` | hybrid |
| | Qwen3-32B | 2025-04 | `Qwen3-32B` | hybrid |
| | QwQ-32B | 2025-03 | `QwQ-32B` | yes |
| | Qwen3 Next 80B A3B Thinking | 2025-09 | `qwen3-next-80b-a3b-thinking` | yes |
| | Qwen3 Next 80B A3B Instruct | 2025-09 | `qwen3-next-80b-a3b-instruct` | no |
| | Qwen3 30B A3B Instruct (2507) | 2025-07 | `qwen3-30b-a3b-instruct-2507` | no |
| | Qwen3 235B A22B (2507) | 2025-07 | `qwen3-235b-a22b-2507` | hybrid |
| | Qwen3 235B A22B Thinking (2507) | 2025-07 | `qwen3-235b-a22b-thinking-2507` | yes |
| Meta | Llama-4 Maverick | 2025-04 | `Llama-4-Maverick-17B-128E-Instruct` | no |
| | Llama-4 Scout | 2025-04 | `Llama-4-Scout-17B-16E-Instruct` | no |
| DeepSeek | DeepSeek-R1-0528 | 2025-05 | `DeepSeek-R1-0528` | yes |
| | DeepSeek-R1 | 2025-01 | `DeepSeek-R1` | yes |
| | DeepSeek-V3 | 2025-03 | `DeepSeek-V3-0324` | no |
| | DeepSeek-V3.1 | 2025-08 | `Deepseek-chat-v3.1` | no |
| | DeepSeek-V3.2-Exp | 2025-09 | `Deepseek-v3.2-exp` | no |
| Zhipu AI | GLM 4.5 | 2025-07 | `glm-4.5` | hybrid |
| | GLM 4.6 | 2025-09 | `glm-4.6` | hybrid |
| OpenAI (OSS) | GPT OSS 20B | 2025-08 | `gpt-oss-20b` | no |
| | GPT OSS 120B | 2025-08 | `gpt-oss-120b` | no |
| MiniMax | MiniMax-M1 | 2025-06 | `MiniMax-M1-80k` | yes |
| Ai2 (AllenAI) | Olmo 3.1 32B Instruct | 2025-11 | `allenai/Olmo-3.1-32B-Instruct` | no |

For each model, we set the temperature as 1. Proprietary models are accessed via their official APIs, while open-source models are served through third-party providers located in US such as Together AI. This setup eliminates the need for local GPU resources. For the o3 and o4-mini models, we set the parameter of `reasoning_effort` as `medium`. For Claude-series models, we enable *extended thinking* with a `budget_tokens` value of 2,048. For Gemini-2.5-series models, we use the default setting where the thinking mode is on and thinking budget is dynamic.

## B.5 Detecting Anomalous Users

We adopt the same approach as Chatbot Arena [14] for detecting anomalous users. Specifically, we consider a dataset consisting of $N$ distinct IPs, and denote the full set of IP addresses as $\text{IP} = \{1, \ldots, N\}$. Suppose a new user provides a sequence of ratings $Y'_1, \ldots, Y'_n$ in response to a sequence of actions $A'_1, \ldots, A'_n$. Our goal is to assess whether this user's rating behavior aligns with the historical distribution of ratings associated with each action.

To formalize this, for a given action $a$, we define the historical ratings set $Y_a = \{Y_t : A_t = a\}$. When the new user submits a rating $Y'_i$ for action $A'_i$, we compute the following statistic:

$$p_i = \frac{\sum_{y \in Y_{A'_i}} \mathbb{1}\{y \geq Y'_i\} + 1}{|Y_{A'_i}| + 1}.$$

Assuming the null hypothesis that $Y_{A'_i}$ and $Y'_i$ are exchangeable, the value $p_i$ serves as a valid $p$-value. The dependence among these $p$-values becomes negligible asymptotically.

To perform sequential testing under this null hypothesis, we apply Fisher's method for combining $p$-values [27], alongside a modified Bonferroni correction. Specifically, after the $j$-th rating from a test user, we calculate:

$$M_j = -2 \sum_{i=1}^{j} \log(p_i).$$

We then assess anomalous behavior at five randomly selected values of $\{1, \ldots, j\}$. A user is flagged as anomalous if $M_j \geq \chi^2_{2j, 1-\alpha/5}$ at any of these selected points.

## C  SciArena Data

### C.1  Annotator Biographies of 102 Researchers Involved in Initial Data Collection

| ID | Position | Field | Publications |
|----|----------|-------|--------------|
| *Engineering* | | | |
| 1 | PhD student | Computer Science | 5-10 |
| 2 | PhD student | Civil Engineering | 5-10 |
| 3 | Graduate student | Optical Engineering | 2-5 |
| 4 | PhD student | Aviation Engineering | 5-10 |
| 5 | PhD student | Computer Science | 5-10 |
| 6 | PhD student | Electrical Engineering | 2-5 |
| 7 | Graduate student | Industrial Design | 2-5 |
| 8 | PhD student | Engineering Management | 5-10 |
| 9 | PhD student | Mechanical Engineering | 5-10 |
| 10 | PhD student | Environment Engineering | 5-10 |
| 11 | Graduate student | Computer Science | >10 |
| 12 | PhD student | Computer Science | 5-10 |
| 13 | PhD student | Electrical Engineering | 5-10 |
| 14 | Graduate student | Power Engineering | 5-10 |
| 15 | PhD student | Environment Engineering | 5-10 |
| 16 | PhD student | Optical Engineering | 5-10 |
| 17 | PhD student | Computer Science | 2-5 |
| 18 | PhD student | Computer Science | 5-10 |
| 19 | Graduate student | Engineering Management | 5-10 |
| 20 | PhD student | Mechanical Engineering | 5-10 |
| 21 | PhD student | Civil Engineering | >10 |
| 22 | PhD student | Industrial Design | 5-10 |
| 23 | Graduate student | Aviation Engineering | 2-5 |
| 24 | PhD student | Computer Science | 5-10 |
| 25 | PhD student | Computer Science | 5-10 |
| 26 | Graduate student | Computer Science | 2-5 |
| 27 | PhD student | Computer Science | >10 |
| 28 | PhD student | Computer Science | 5-10 |
| 29 | PhD student | Computer Science | 2-5 |
| 30 | Graduate student | Computer Science | 2-5 |
| 31 | PhD student | Power Engineering | 5-10 |
| 32 | PhD student | Electrical Engineering | 5-10 |
| 33 | PhD student | Power Engineering | 2-5 |
| 34 | PhD student | Optical Engineering | 5-10 |
| 35 | PhD student | Environment Engineering | >10 |
| 36 | PhD student | Engineering Management | 2-5 |
| 37 | PhD student | Mechanical Engineering | 5-10 |
| 38 | PhD student | Civil Engineering | 5-10 |
| 39 | Graduate student | Industrial Design | >10 |
| 40 | Graduate student | Aviation Engineering | 5-10 |
| 41 | Graduate student | Computer Science | 5-10 |
| 42 | Graduate student | Computer Science | 5-10 |
| *Healthcare* | | | |
| 43 | PhD student | Clinical Medicine | 5-10 |
| 44 | Graduate student | Pharmacy | 2-5 |
| 45 | PhD student | Basic Medicine | 5-10 |
| 46 | PhD student | Public Health | 5-10 |
| 47 | Graduate student | Neuroscience | >10 |
| 48 | PhD student | Biomedical Engineering | 2-5 |
| 49 | PhD student | Clinical Medicine | 5-10 |
| 50 | PhD student | Pharmacy | 5-10 |
| 51 | Graduate student | Basic Medicine | 5-10 |
| 52 | PhD student | Public Health | 5-10 |

*Continued on next page*

| ID | Position | Field | Publications |
|---|---|---|---|
| 53 | Graduate student | Neuroscience | 2-5 |
| 54 | PhD student | Biomedical Engineering | 5-10 |
| 55 | PhD student | Clinical Medicine | 2-5 |
| 56 | Graduate student | Public Health | 5-10 |
| 57 | PhD student | Basic Medicine | 2-5 |
| 58 | Graduate student | Pharmacy | 2-5 |

### *Natural Sciences*

| ID | Position | Field | Publications |
|---|---|---|---|
| 59 | PhD student | Environmental Science | 5-10 |
| 60 | Graduate student | Animal Science | 5-10 |
| 61 | PhD student | Ecology | 2-5 |
| 62 | PhD student | Crop Science | 5-10 |
| 63 | Graduate student | Chemistry | 2-5 |
| 64 | PhD student | Horticulture | 5-10 |
| 65 | PhD student | Food Science | 2-5 |
| 66 | Graduate student | Geographic Information Science | 5-10 |
| 67 | PhD student | Material Science | 5-10 |
| 68 | PhD student | Mathematics | 2-5 |
| 69 | Graduate student | Physics | 2-5 |
| 70 | PhD student | Biology | 2-5 |
| 71 | PhD student | Environmental Science | 2-5 |
| 72 | Graduate student | Animal Science | 5-10 |
| 73 | PhD student | Ecology | 5-10 |
| 74 | PhD student | Crop Science | 5-10 |
| 75 | Graduate student | Chemistry | 5-10 |
| 76 | PhD student | Horticulture | >10 |
| 77 | PhD student | Food Science | 5-10 |
| 78 | Graduate student | Geographic Information Science | 5-10 |
| 79 | PhD student | Material Science | 2-5 |
| 80 | PhD student | Mathematics | 2-5 |
| 81 | Graduate student | Physics | 2-5 |
| 82 | PhD student | Biology | 5-10 |
| 83 | PhD student | Ecology | 2-5 |
| 84 | Graduate student | Material Science | 2-5 |
| 85 | PhD student | Biology | 5-10 |
| 86 | Graduate student | Chemistry | 2-5 |

### *Humanities & Social Sciences*

| ID | Position | Field | Publications |
|---|---|---|---|
| 87 | PhD student | Management | 5-10 |
| 88 | Graduate student | Arts and Media | 2-5 |
| 89 | PhD student | Economics | 2-5 |
| 90 | PhD student | Linguistics | 5-10 |
| 91 | Graduate student | Psychology | 2-5 |
| 92 | PhD student | History | 5-10 |
| 93 | Graduate student | Management | 5-10 |
| 94 | PhD student | Arts and Media | 2-5 |
| 95 | PhD student | Economics | 5-10 |
| 96 | Graduate student | Linguistics | 5-10 |
| 97 | PhD student | Psychology | 2-5 |
| 98 | PhD student | History | 2-5 |
| 99 | Graduate student | Management | 2-5 |
| 100 | PhD student | Economics | 5-10 |
| 101 | Graduate student | Psychology | 5-10 |
| 102 | Graduate student | Management | 2-5 |

## C.2 SciArena Data Analysis

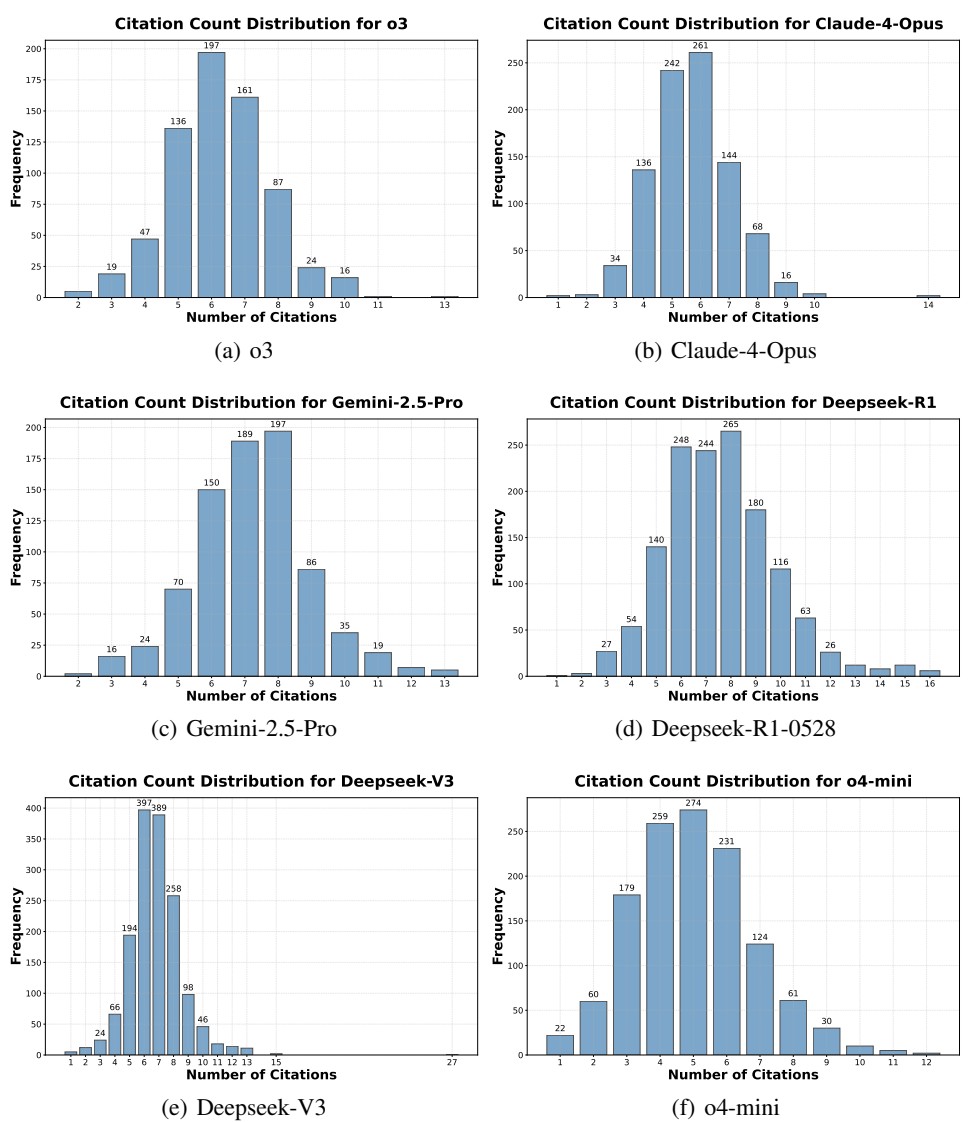

Figure 6: Citation count distribution for different models (Part 2).

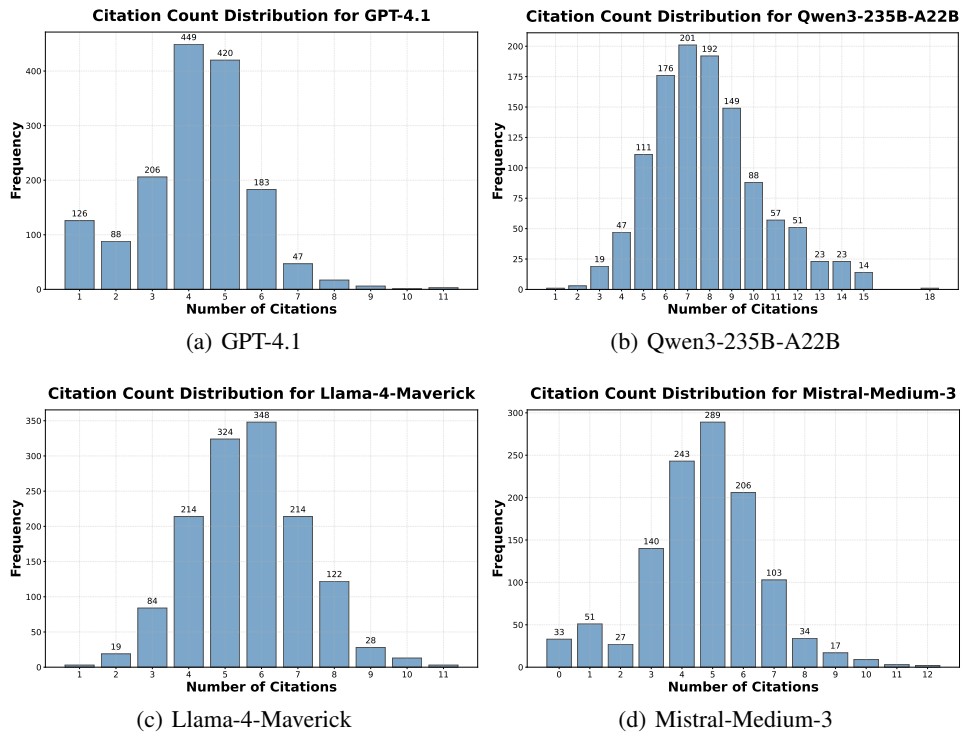

Figure 7: Citation count distribution for different models (Part 2).

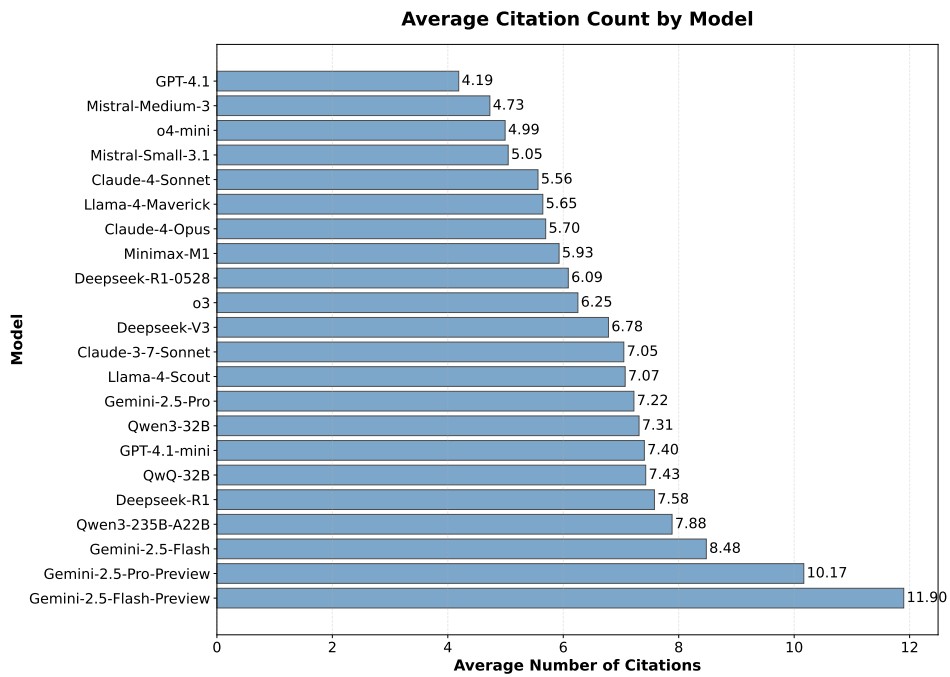

Figure 8: Average Citation Count for selected models.

## C.3 Prompts for Question Category Classification

Figure 9: The prompt used for question category classification.

# D SciArena Leaderboard Results

## D.1 In-depth Analysis

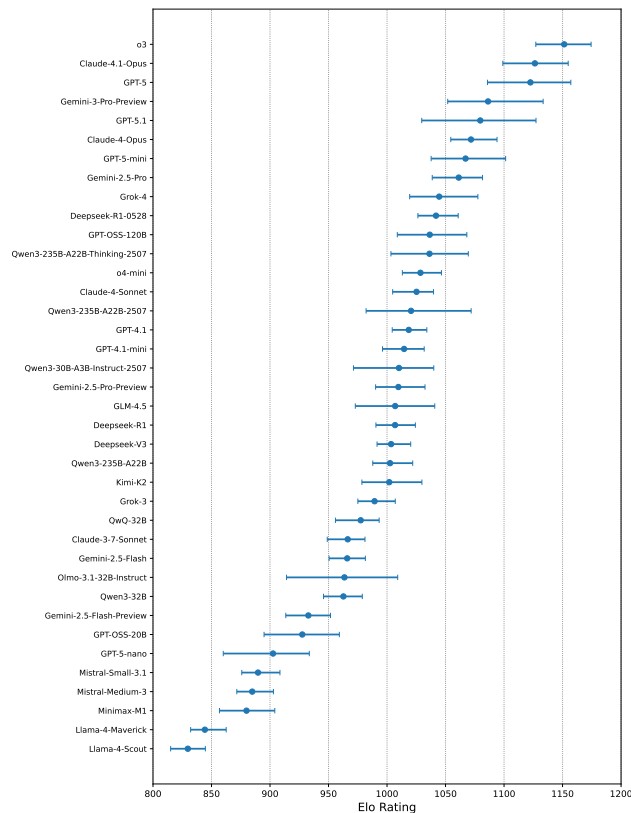

Figure: Leaderboard showing Elo ratings with 95% confidence intervals.

Figure 10: Leaderboard from SciArena, showing Elo ratings with 95% confidence intervals.

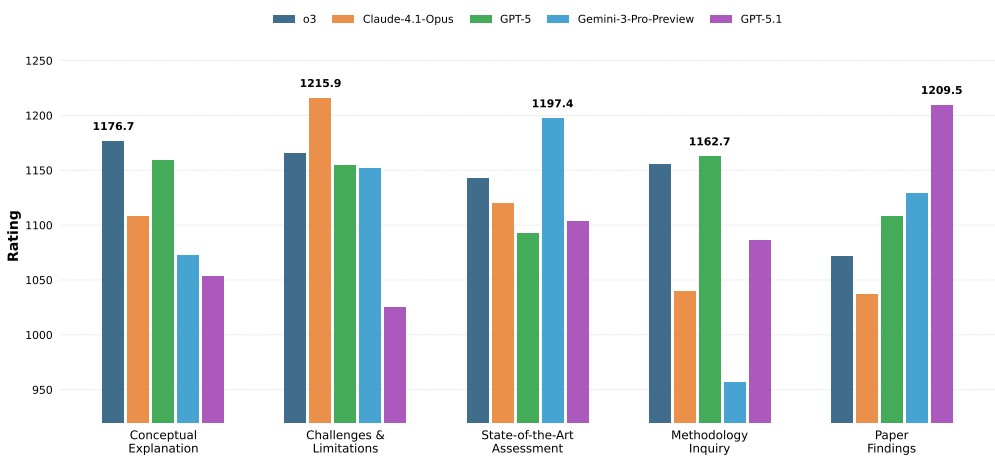

Figure 11: Elo ratings of top-5 models across question categories.

## D.2 Analysis of Citation Features in SciArena Evaluation

To systematically investigate the impact of citation-related features on user preferences in SciArena, we randomly sample 3,000 voting instances for analysis. Each citation within the responses is labeled using the o4-mini model to determine its attribution category: *supporting*, *irrelevant*, or *contradicting*. The prompt used for this classification is shown in Figure 12. We merge the *irrelevant* and *contradicting* categories for downstream analysis. For each response, we count the number of (1) supporting citations and (2) irrelevant or contradicting citations. These counts are then used as features in a Bradley-Terry model, consistent with the methodology applied in other control experiments.

---

**[System Prompt]**
You are a professional literature citation analysis expert who needs to judge the relationship between citation content and text statements.

**[User Prompt]**
Please analyze whether the following citation is consistent with the content mentioned in the response.

Response content:{response}
Citation author: {concise_authors}
Citation original content:{content}

Please judge whether the citation's original content is consistent with the statements in the response based on the specific content mentioned about "{concise_authors}" in the response.

Special attention:
1. Please pay attention to the letters at the end of concise_authors (such as a, b, c, d, etc.) to ensure correct citation matching
2. Carefully compare the specific statements about this author in the response with the citation's original content
3. Classification criteria:
- supporting: The citation content supports the statements in the response
- irrelevant: The citation content is unrelated or irrelevant to the statements in the response
- contradicting: The citation content contradicts the statements in the response

Please classify based on the above analysis and provide a brief explanation.

---

Figure 12: The prompt used for citation attribution classification.

## D.3 Analysis of o3 Model

### D.3.1 Examples Illustrating o3's Detailed Elaboration on Cited Papers

---

**Detailed Elaboration on Cited Paper (Example 1)**

**User question:** What statistical models effectively capture polysemy development and lexical network growth?

**Caude-Opus-4 Response:**
(...abbreviated...) Parallel to network growth, statistical analyses reveal clear developmental patterns in polysemy. The mean polysemy of words used by children exhibits a two-phase growth pattern, characterized by a rapid increase that slows after a breakpoint at approximately 31 months of age (Casas et al. 2016a). (...abbreviated...)

**o3 Response:**
(...abbreviated...) Developmentally, longitudinal CHILDES analyses that compute the average number of WordNet or SemCor senses per produced token reveal a two-phase "fast–slow" rise in children's mean polysemy with a breakpoint near 32 months, indicating that polysemy acquisition follows a quantifiable growth curve (Casas et al. 2016a). (...abbreviated...)

**Analysis:** Compared with the Caude-Opus-4 version, o3 explicitly spells out that mean polysemy was measured as the average number of WordNet or SemCor senses per token rather than just "mean polysemy." It labels the developmental pattern as a two-phase "fast–slow" rise, giving the growth curve a precise conceptual frame. By framing polysemy acquisition as a quantifiable growth trajectory, o3 highlights both the method and the empirical milestone. This extra methodological detail and clear terminology make o3's answer a more detailed elaboration on the cited paper.

---

**Detailed Elaboration on Cited Paper (Example 2)**

**User question:** How have benchmarks evolved to assess language model performance in lesser-studied academic disciplines?

**Caude-Opus-4 Response:**
(...abbreviated...) As models achieve near-saturation performance on existing benchmarks, researchers have responded by creating more challenging, domain-specific evaluations that test deeper reasoning within particular fields rather than broad surface-level knowledge (Pricope et al. 2025a). (...abbreviated...)

**o3 Response:**
(...abbreviated...) HardML extends the MMLU philosophy but contributes hundreds of fresh questions spanning algorithm design, statistical inference, and practical ML engineering to probe depth rather than breadth in DS/ML (Pricope et al. 2025a). (...abbreviated...)

**Analysis:**
Compared to Caude-Opus-4's high-level summary, o3 dives deeper into Pricope et al. (2025a) by enumerating HardML's key question categories—algorithm design, statistical inference, and practical ML engineering, which provides more detailed elaboration.

### D.3.2 Examples Illustrating o3's More Professional & Precise Terminology

> **More Professional&Precise Terminology**
>
> **User question:** What are the main challenges and solutions related to maintaining model performance when using low-bit attention techniques during training?
>
> **Caude-Opus-4 Response:**
> Low-bit attention techniques during training face significant challenges in maintaining model performance, primarily due to quantization-induced information loss and training instability. **The fundamental challenge stems from the unique properties of transformers' attention mechanisms, which exhibit high quantization sensitivity compared to convolutional neural networks. As Huang et al. demonstrate, transformers display variation behaviors across three hierarchies: module-level quantization sensitivities, outliers in weight and activation distributions, and oscillation in dynamic parameter fluctuations**, all of which contribute to training instability and performance degradation (Huang et al. 2023a).
> **The presence of outliers in attention computations poses a particularly severe challenge, as these values expand the quantization range and make low-bit quantization difficult, with some approaches showing over 10% accuracy degradation when quantizing to sub-2-bit representations (Du et al. 2024a).** Additionally, the computational pipeline of quantization methods presents efficiency challenges, as most quantized training methods adopt a quantize-compute-dequantize paradigm that proves particularly inefficient for transformer architectures due to frequent data conversion overhead (Hao et al. 2025a). (...abbreviated...)
>
> **o3 Response:**
> Training transformers with low-bit attention confronts several intertwined problems. **Outliers in key/value activations and in the soft-max scores inflate the dynamic range, so rounding to 4 bits or less causes frequent overflow/underflow and sharp accuracy loss (Wang et al. 2025b). Quantization sensitivity also varies widely across attention sub-modules; query/key projections, soft-max, and residual connections oscillate differently during optimization, making low-bit training unstable and hampering convergence (Huang et al. 2023a).** Moreover, low-precision formats are not natively supported for optimizer states or inter-GPU communication, so gradients and moments must be promoted, negating speed benefits unless additional engineering is introduced (Xie et al. 2024a). Finally, aggressive rounding can destabilize Adam-style updates, leading to divergence unless the optimizer itself is re-designed for few-bit arithmetic (Huang et al. 2025a).
> Recent studies present complementary solutions. Variation-aware quantization assigns module-specific scale factors, detects outliers with bin regularization, and distills knowledge from a full-precision teacher, recovering 2-bit transformer performance on both language and vision tasks (Huang et al. 2023a). Stable-SPAM redesigns Adam to store moments in 4-bit integers with norm-aware clipping, enabling 4-bit training that is even more stable than standard 16-bit baselines (Huang et al. 2025a). (...abbreviated...)

> **Analysis:** o3's response is more professional and precise because first it explicitly breaks down the "sub-modules" of attention—"query/key projections, soft-max, and residual connections"—showing a deeper technical understanding. It references the concrete architectural elements affected by quantization, rather than just stating "three hierarchies." This terminology is more specific and demonstrates expertise in the transformer architecture. Besides, o3 is more precise by specifying where the outliers occur ("key/value activations and in the soft-max scores") and the concrete effect ("inflate the dynamic range," "frequent overflow/underflow"), which are technical terms in quantization.

### D.3.3 Examples Illustrating o3's Clear Structured Presentation

---

**Clear Structured Presentation**

**User question:** What are the main challenges and solutions for achieving controllable video generation across diverse input conditions?

**Gemini-2.5-Pro:**

**A primary challenge** in controllable video generation is the fragmented landscape of specialized models, which often lack generalizability across different control types (Zhang et al. 2025a). This makes it difficult to adapt existing spatial controls from images, as frameworks like Control-Net cannot be directly integrated into new video backbones due to feature mismatches (Lin et al. 2024a).

Furthermore, many methods require impractical, dense condition sequences for every frame to maintain temporal consistency (Wang et al. 2024c). To overcome these issues, researchers propose unified frameworks using efficient adapters, which add diverse controls such as sketches or depth maps to pre-trained models with minimal retraining (Wang et al. 2024b). Other solutions feature a unified pipeline to encode various signals into a common representation, enabling a single generator to handle multiple control types (Zhang et al. 2025a).

**A separate significant challenge** is the limited capacity of diffusion models to interpret complex, heterogeneous multimodal inputs simultaneously, which can cause conflicts between conditions and degrade video quality (Chen et al. 2024b). To address this, frameworks like Any2Caption decouple condition interpretation from synthesis, using a Multimodal Large Language Model (MLLM) to translate diverse inputs into a single, dense text caption that provides clearer guidance to the video generator (Wu et al. 2025b).

**o3 Response:**

Controllable video generation must reconcile heterogeneous user inputs with high visual fidelity and temporal coherence. **Key technical hurdles include (i) the feature-space mismatch that prevents simply porting 2D control modules such as ControlNet to video backbones, hampering reuse across models (Lin et al. 2024a); (ii) dependence on dense, frame-by-frame control sequences that place an impractical burden on creators and still struggle to propagate signals through time (Wang et al. 2024d); (iii) the limited capacity of conventional text encoders to interpret complex, multimodal intent, causing weak alignment between the specified constraints and the produced clip (Wu et al. 2025b); and (iv) geometric and temporal inconsistency when only 2D cues are available, especially for camera or object motion, because the underlying 3D scene is ignored (Gu et al. 2025a).**

Recent research proposes complementary remedies. EasyControl attaches a lightweight condition adapter to pretrained text-to-video diffusion models and propagates a single sparse map through residual summation, yielding plug-and-play control over images, edges, depth, sketches and masks with minimal extra training (Wang et al. 2024b). VCtrl generalises this idea via a unified control encoder and sparse residual connections, enabling fine-grained, multi-task manipulation while keeping the original generator frozen (Zhang et al. 2025a). (...abbreviated...)

---

**Analysis:** o3 directly enumerate the "main challenges" using (i) - (iv), and then introduce the "solutions" separately in paragraphs. This itemized list makes it clear to readers at a glance. In contrast, gemini-2.5-pro only lists challenges and solutions in a streamlined manner, lacking hierarchical markers and making it difficult to quickly grasp the key points.

### D.3.4 Examples Illustrating o3's More Comprehensive Coverage

---

More Comprehensive Coverage

**User question:** What are the most widely adopted open-source tools for time series load forecasting in cloud systems?

**Gemini-2.5-Flash Response:**

While the literature extensively discusses various machine learning and deep learning models for time series load forecasting in cloud systems, such as ARIMA, LSTM, and SVM, explicit mentions of widely adopted open-source tools or frameworks specifically for this task are less prevalent (Mahbub et al. 2024a).

However, some systems designed to automate or facilitate time series forecasting, often leveraging open-source components, have been proposed. AutoAI-TS, for instance, is presented as a zero-configuration system framework designed to automate the time series forecasting process, incorporating models from the open-source domain and using scikit-learn compatible APIs (Shah et al. 2021a). This system aims to achieve high accuracy with minimal user input by transparently handling tasks like feature engineering, training, parameter tuning, and model ranking (Shah et al. 2021a).

Another relevant system is Castor, a cloud-native platform for managing time series data and models, which integrates models from open-source domains and leverages serverless technology (O'Donncha et al. 2021a). While Castor has been applied in specific domains like renewable energy, its design as a general system for time series data management suggests applicability to cloud load forecasting (O'Donncha et al. 2021a). Additionally, cloud-native time-series databases like Lindorm TSDB offer integrated machine learning functions, including time series forecasting algorithms directly accessible via SQL, designed for large-scale monitoring systems in distributed environments (Shen et al. 2023a). These platforms highlight a trend towards integrated or automated solutions that utilize underlying open-source algorithms and frameworks to address the complexities of cloud workload prediction.

**o3 Response:**

The landscape of open-source tooling for time-series load forecasting in cloud settings can be traced back to the Network Weather Service (NWS), a long-standing, freely available middleware that is still routinely used to obtain short-term CPU and bandwidth forecasts for grid and cloud schedulers (Hu et al. 2014a).

At the model-development layer, the de-facto standard today is the Python scientific stack: scikit-learn, statsmodels and related libraries are packaged behind higher-level automation services such as AutoAI-TS, which provides a zero-configuration interface that automatically builds, tunes and ranks ARIMA, gradient-boosting and deep-learning pipelines while remaining fully compatible with the scikit-learn open-source API (Shah et al. 2021a).

For production deployment, cloud operators increasingly rely on cloud-native platforms that integrate storage, training and inference. Castor, released under an Apache-style licence, exemplifies this approach by coupling serverless IBM Cloud Functions with an ingestion layer to support recurrent retraining and multi-model comparison for workload prediction in aquaculture and other IoT-driven workloads (O'Donncha et al. 2021a). On the data-management side, Alibaba's Lindorm TSDB embeds anomaly detection and forecasting primitives directly into its SQL engine, allowing operations teams to invoke ARIMA or Prophet-like models without moving data out of the open-source database (Shen et al. 2023a).

Finally, modern deep-learning research projects such as Informer publish reference implementations that are widely reused by cloud-scaling studies. Ding et al. demonstrate that the open-source Informer codebase can outperform classical ARIMA, RNN and LSTM baselines when predicting container-level CPU demand, making it a popular choice for academic and industrial prototypes (Ding et al. 2025a).

Together, these tools—NWS for lightweight signal collection, scikit-learn/statsmodels pipelines automated by AutoAI-TS, cloud-native platforms like Castor and Lindorm TSDB, and reference deep-learning frameworks such as Informer—constitute the most commonly adopted open-source stack for load forecasting in cloud systems today.

---

**Analysis:** o3 introduces work with a wider time span (2014-2025) and includes the latest Informer (Ding et al. 2025a). It also covers more research and analyzes them comprehensively; The other response was last cited by Shen et al. 2023a and did not mention any significant developments in 2024 or beyond, nor did it mention any new frameworks for deep learning.

### D.4 Model Failure Case Analysis

### D.4.1 Examples Illustrating Errors of Failure to Answer the Question

> **Failure to Answer the Question**
>
> **User question:**
> How is precision nutrition transforming contemporary poultry feed formulation research?
>
> **Gemini-2.5-Flash-Preview Response:**
> Precision nutrition is transforming poultry feed formulation research by shifting from traditional static or multi-phase feeding systems to precisely matching the daily nutrient requirements of birds (Moss et al. 2021f, Moss et al. 2021e). This practice aims to ensure the dietary nutrient supply equals the animal's requirement, thereby minimizing periods of under and over-feeding (Moss et al. 2021f, Cambra-López et al. 2022c).
> This paradigm shift is anticipated to improve production efficiency, reduce the significant feed costs in poultry production (Moss et al. 2021a, Adebayo et al. 2019a), and enhance sustainability by minimizing nutrient excretion and environmental impact (Pomar et al. 2022a, Bist et al. 2024a). Implementing precision nutrition relies heavily on advanced technologies, including precision feeding systems that enable accurate feed delivery and blending on-farm (Moss et al. 2021b, Moss et al. 2021d, Moss et al. 2021f), and sophisticated computational approaches like mathematical models, artificial intelligence, and the use of big data to estimate dynamic requirements and optimize formulations (Unknown et al. 2024a, Pomar et al. 2022a, Kogut et al. 2022a, Moss et al. 2021g).
> Research is also delving into areas like the role of functional amino acids, biomarkers, and diet-microbiota interactions to further refine precision strategies (Rochell et al. 2023a, Cambra-López et al. 2022a, Kogut et al. 2022a). The ability to make rapid adjustments based on performance, environmental changes, or farm-specific needs represents a key advantage of this approach (Moss et al. 2021c).
>
> **References:**
> 1. A. Moss, P. Chrystal, D. Cadogan, S. Wilkinson, T. Crowley, M. Choct, Precision feeding and precision nutrition: a paradigm shift in broiler feed formulation? 2021.
> 2. C. Pomar, A. Remus, 242 The Impact of Feed Formulation and Feeding Methods on Pig and Poultry Production on the Environment. 2022.
> 3. Exploring the potential of artificial intelligence in feed formulation to advance poultry health and One-Health.
> 4. M. Cambra-López, P. Marín-García, C. Lledó, A. Cerisuelo, J. J. Pascual, Biomarkers and De Novo Protein Design Can Improve Precise Amino Acid Nutrition in Broilers. 2022.
> 5. S. J. Rochell, 80 Precision Nutrition to Enhance Poultry Performance and Health: the Role of Functional Amino Acids. 2023.
> 6. R. Bist, Keshav Bist, S. Poudel, Deepak Subedi, Xiao Yang, Bidur Paneru, Sudhagar Mani, Dongyi Wang, Lilong Chai, Sustainable poultry farming practices: a critical review of current strategies and future prospects. 2024.
> 7. Oluwadare Samuel Adebayo, Gabriel Arome Junior, Ogunrinde Oluwakemi Grace, Tabu-Genetic Algorithm-Based Model for Poultry Feed Formulation. 2019.
> 8. M. Kogut, Role of diet-microbiota interactions in precision nutrition of the chicken: facts, gaps, and new concepts. 2022.

> **Error Type:** Failure to Answer the Question
> **Explanation:** The question is: "How is precision nutrition transformed contemporary poultry feed-formulation research?" What must be addressed is how the research objectives have shifted, how the tools, methodologies and experimental designs have evolved, and how the entire workflow of data collection, model building and formula validation has been reshaped. However, the current answer focuses mainly on the advantages of precision nutrition, and on the technologies used to implement it, such as precision feeders and AI, without clearly articulating how the research paradigm itself has changed.

### D.4.2 Examples Illustrating Errors of Conflict with Cited Papers

---

**Conflict with Cited Papers**

**User question:**
During exceptional circumstances, such as the pandemic, does collaboration among researchers tend to expand existing partnerships or form new ones?

**Llama-4-Scout Response:**
The COVID-19 pandemic has presented unprecedented challenges to researchers worldwide, affecting their collaboration patterns. Studies suggest that during exceptional circumstances like the pandemic, collaboration among researchers tends to expand existing partnerships rather than form new ones (Alshayhan et al. 2023a). Emergency management leaders leverage established partnerships to bring in new partners with needed expertise, adjusting based on new information and meeting specific needs (Alshayhan et al. 2023a).

However, some research indicates that the pandemic also facilitated the formation of new partnerships, particularly among early-career researchers and those from different countries (Brown et al. 2022a, López-Vergès et al. 2021a). For instance, a study found that junior researchers showed a significant surge in participation and collaboration during the pandemic, with their network dynamics becoming tighter (Ahrabian et al. 2024f). Additionally, researchers from the Global South and North collaborated more closely during the pandemic, with North-South partnerships helping to navigate challenges and achieve greater engagement and impact (Yorke et al. 2023a).

The pandemic has also accelerated scientific activity, with researchers adapting to new circumstances and finding ways to collaborate effectively despite physical distancing measures (Ahrabian et al. 2024a). Overall, the pandemic has highlighted the resilience and adaptability of the scientific community, with collaboration patterns evolving to meet the challenges posed by the crisis (Ahrabian et al. 2024a). While existing partnerships have been leveraged, new ones have also been formed, particularly among researchers with similar interests and goals (Brown et al. 2022a).

**Example Citation Content from Alshayhan et al. 2023a** Emergency management is a key government function for mitigating risks and reducing the impacts of disasters. Emergency management leaders play a critical role in preparing for and responding to disasters whose impacts are exacerbated by a pandemic. Using the example of the compound threat of hurricanes and the COVID-19 pandemic, this qualitative research uses insights from emergency management professionals to describe collaborative approaches and leadership skills that help balance the needs for stability and flexibility. Data collected using focus groups and one-on-one interviews with emergency management professionals highlight that collaboration involved existing and new partners in a changing and uncertain environment that challenged traditional leadership of emergency management.

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

### D.4.3 Examples Illustrating Errors of Lack of Details

---

**Lack of Detail**

**User question:**
What innovative literary devices symbolize technological advancement in cyberpunk narratives?

**Llama-4-Maverick Response:**
Cyberpunk narratives often employ innovative literary devices to symbolize technological advancement, reflecting the genre's focus on the intersection of technology and humanity. One such device is the use of prosthetic enhancements, which serve as a literal manifestation of posthumanism and its proposed hybridized politics (Collins et al. 2014a).

William Gibson's works, for example, feature characters with technologically enhanced bodies, such as Molly Millions, who has 'surgical inlays sealing her eyes in their sockets' and 'ten blades snickerstraight out from their recesses beneath her nails' (Collins et al. 2014c). These enhancements blur the lines between human and machine, illustrating the complex relationship between technology and the body. The use of virtual reality constructs and simulated experiences also serves to further complicate this relationship, creating a sense of 'hyperreality' that challenges the notion of a clear distinction between the physical and digital worlds (Zaidi et al. 2020a). By employing these devices, cyberpunk narratives are able to explore the implications of technological advancement on human identity and society, often presenting a dystopian vision of a future where technology has become a dominant force. This vision is characteristic of the cyberpunk genre, which frequently critiques the consequences of unchecked technological progress.

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

---

**Error Type:** Lack of Detail
**Explanation:** The response suffers from a lack of detail: it gestures at prosthetic enhancements and "hyperreality" without showing how these devices actually operate within the plot or convey thematic meaning, and it leans on a single example (Molly Millions) rather than examining multiple characters, scenes, or texts. It never unpacks the narrative mechanics by which body augmentations or virtual constructs influence reader engagement or reflect posthuman concerns. There is no contextualization in broader theoretical or cultural frameworks—no reference to posthumanist theory, Lo-Tek resistance, or genre evolution—nor any comparative analysis across authors or works. Finally, the response offers no quantification or formal grounding (such as counts of device occurrences or structural modeling) to substantiate its assertions.

### D.4.4 Examples Illustrating Errors of Misunderstanding Terminology

---

**Misunderstanding of Terminology**

**User question:**
What is the difference between LCOE and LCOA?

**Claude-3-7-Sonnet Response:**
The Levelized Cost of Electricity (LCOE) and the Levelized Cost of Ammonia (LCOA) are economic metrics used in energy systems analysis, but with distinct focuses. LCOE represents the average cost per unit of electricity generated over a project's lifetime, calculated by dividing the total lifecycle costs by the total electricity produced (Gattie et al. 2023a). It serves as a standardized framework for comparing the cost-effectiveness of diverse electricity generation technologies (Kafando et al. 2024a). The LCOE formula typically includes capital investment, operation and maintenance expenditures, fuel costs, and accounts for the time value of money using a discount rate (Emblemsvåg et al. 2023a).

In contrast, LCOA specifically measures the cost of producing ammonia, calculated as the ratio of production costs to the mass of ammonia produced (Nayak-Luke et al. 2018a). While LCOE focuses on electricity generation costs, LCOA encompasses the entire ammonia production process, including hydrogen generation (often via electrolysis), air separation units, and the Haber-Bosch process (Nayak-Luke et al. 2020a). The relationship between these metrics is significant since electricity costs (represented by LCOE) often constitute a major component of the LCOA in green ammonia production systems (Pan et al. 2023a). Analysis shows that changes in LCOE directly impact LCOA; for instance, a change of ±0.89 GBP/MWh in LCOE results in a change of ±10 GBP/tonne in LCOA (Nayak-Luke et al. 2018a). This relationship becomes particularly important when evaluating renewable-powered ammonia production systems, where electricity generation represents a significant portion of the total costs.

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

---

**Error Type:** Misunderstanding of Terminology
**Explanation:** In this discussion, LCOA means Levelized Cost of Abatement: the discounted lifetime cost per tonne of CO2e avoided, stated in USD/tCO2e. People use it to compare the cost-effectiveness of decarbonization options, not ammonia production. The earlier answer wrongly treated LCOA as Levelized Cost of Ammonia, which distorted the comparison with LCOE.

### D.4.5 Examples Illustrating Errors of Incoherent Structure

**User question:**
What are the optimal plant combinations for different types of wastewater treatment, and what are the recent research advancements or practical guidelines in this field?

**Deepseek-R1 Response:**
Recent research highlights diverse plant combinations and system designs optimized for specific wastewater types. For constructed wetlands (CWs), ornamental species like Anthurium spp., Canna hybrids, and Typha dominguensis effectively remove contaminants in municipal wastewater when paired with PET filter media, achieving 85.2% total phosphorus removal (Herazo et al., 2023b).
Hydroponic systems, particularly misting (MT) and aeroponic (AP) configurations, enhance oxygen availability in the rhizosphere, promoting plant growth and organic pollutant removal, with rice experiments showing improved dissolved oxygen dynamics (Clyde-Smith et al., 2023a). Duckweed (Lemnaceae) integrated with anaerobic digestion in multi-trophic systems demonstrates dual benefits in dairy wastewater treatment and biomass production for aquaculture feed (Oláh et al., 2023a). For heavy metals, Salvinia molesta and Pistia stratiotes show promise in phytoremediation, leveraging root exudates and microbial interactions (Fonseka et al., 2023a).
Floating treatment wetlands (FTWs) combining Phragmites australis (fast-growing) and Juncus effusus (slow-growing) balance rapid pollutant uptake and long-term stability, especially in nutrient-rich environments (Arslan et al., 2023a). Recent advancements emphasize microbial synergies: Eichhornia crassipes paired with bacterial consortia achieves 98.83% chromium removal (Rashid et al., 2023a), while co-culturing microalgae (Chlamydomonas reinhardtii) with yeast (Saccharomyces cerevisiae) optimizes TOC (82%) and phosphate (94%) removal (Takahashi et al., 2024a). Practical guidelines stress selecting plants based on influent characteristics (e.g., Ipomoea aquatica for phosphorus-rich wastewater (Asih et al., 2024a)) and local climate adaptability. System design considerations include hydraulic retention time, substrate choice (e.g., PET media (Herazo et al., 2023b)), and integrating microbial communities to enhance phytoremediation efficiency (Riva et al., 2024a). These strategies align with low-cost, energy-efficient solutions for diverse wastewater challenges.

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

---

**Error Type:** Incoherent Structure
**Explanation:** The review suffers from the absence of a clear narrative structure: data are presented as a disconnected catalogue of plant species, removal rates and experimental settings, leaving the reader unsure why particular species are chosen or in which wastewater scenarios they perform best. Because it merely lists outcomes, the discussion lacks analytical depth, offering little insight into the underlying removal mechanisms, operational constraints, performance limits or real-world applicability. Moreover, it fails to synthesise similarities and differences across studies, identify overarching trends or highlight research gaps, making it difficult to extract coherent design principles or future directions.

# E SciArena-Eval Meta-Evaluation Experiment

## E.1 Evaluation Prompts

---

**[System Prompt]**
You are an assistant specialized in processing citation-attributed response. Your task is to transform the review into a structured format by doing the following:

- Ensure all in-text citations in the main text follow the numeric bracket style, e.g. '[1]'.
- Ensure all the brackets (citation) appear at the end of the sentence instead of in the middle of a sentence or come after a clause with comma. e.g. "Guo et al. [1] introduced IdeaBench, a benchmark specifically designed to assess the capability of various LLMs in generating innovative research ideas." should be changed to "Guo et al. introduced IdeaBench, a benchmark specifically designed to assess the capability of various LLMs in generating innovative research ideas [1]."
- Remove any Markdown formatting from the text.
- Make only minimal necessary modifications to achieve this format.

Structure the response in the following format:
{
"references": [], // list of int, each int is the id of the reference in the response.
"response": // Main text of the generated response.
}

Follow the instruction to process the model response.

**[System Prompt]**
You are an expert in classifying questions into categories. Given a question, you should classify each question into a specific category. If there is no suitable category, you should respond with 6 to refer as Others.

Question Categories:
1. Conceptual Explanation: questions about explanations of scientific principles or underlying mechanisms. For example: How does DNA methylation regulate gene expression during fruit ripening?
2. Methodology Inquiry: questions about technical methods, modeling, or simulations. For example: What are the forms and methods of film and game integration?
3. State-of-the-Art Assessment:questions about the current innovations, trends, or breakthroughs in a specific field. For example: How are circular economy principles contributing to industrial emissions reduction lately?
4. Challenges & Limitations: questions about challenges, limitations, or potential areas for improvement. For example: What are the primary challenges hindering the widespread application of uranium based metal organic frameworks?
5. Paper Finding: questions about literature summaries, evidence synthesis, or specific paper retrieval. For example: Research on Legal Issues Involved in the Development and Use of Intellectual Property Information Resources
6. Others: questions not fitting any of the above categories.

Respond with ONLY the category number, nothing else. You should respond with a number ranging from 1 to 6.

You are an expert in scientific literature synthesis. Your task is to evaluate the quality of two AI-generated citation-attributed responses to a user's question. Assess both responses for relevance, accuracy, clarity, and appropriate use of citations. Then, select the response, Output (a) or Output (b), that best address the user's question.

User Question:
{user-query}

Output (a):
{model-a-response}

Output (b):
{model-b-response}

Which is best, Output (a) or Output (b)?

---

Figure 13: The prompt used for pairwise comparison, which is adopted from AlpacaEval [38].

