# OpenReview forum: "SciArena: An Open Evaluation Platform for Non-Verifiable Scientific Literature-Grounded Tasks"
_NeurIPS.cc/2025/Datasets_and_Benchmarks_Track — NeurIPS 2025 Datasets and Benchmarks Track spotlight_

### Official Review · Reviewer_Vq6h · 2025-07-02

**Rating:** 6
**Confidence:** 5

**Summary:**

This paper introduces a new chatbot arena-style platform designed for evaluating LLMs on scientific literature task. The platform integrates a literature retrieval system to retrieve the relevant literature based on user questions, and employs two randomly sampled LLMs to generate citation-grounded responses. The users then vote for the better one.

The authors collected over 8,000 voting examples spanning four core scientific disciplines. They conducted a detailed analysis of the data, revealing interesting patterns such as frequently asked question types.The dataset demonstrates strong reliability, as evidenced by high self-consistency and IAA scores. Using this data, the authors construct an Elo-based leaderboard to assess and compare LLM performance on scientific literature task. The provided preference analysis and case study are insightful.

The paper also introduces a meta-evaluation benchmark based on the collected voting data. The experiment shows that even the best-performing LLM-as-Judge system significantly underperform compared to human judgment.

**Dataset Code Accessibility:**

Yes

**Ethical Considerations:**

No, there are no or only very minor ethics concerns

**Final Justification:**

The new analysis on citation features is interesting and further supports the high quality of the collected data. I believe adding this to the main sections could offer valuable insights for future researchers.
I have also reviewed the other reviewers’ comments and the authors' response. I appreciate the team's continued efforts in maintaining and updating the evaluation platform, such as including the latest models and deep research systems. This makes the work more valuable to the research community.
I share Reviewer kkAU’s view that this is one of the strongest submissions I have reviewed this year. It sets a high standard for the DB track. Therefore, I am raising my score to 6.

**Limitations Weaknesses:**

This paper makes a valuable contribution to the community, and I do not see any major weaknesses. I list some suggestions for further improvement:

- In Section 5.2, the work analyzes response length bias in evaluation and shows that this bias is smaller compared to previous arena-style platforms. However, for literature review generation, citation-related features like number of citations within response, can also bias user preference. Analyzing such features could be insightful
- The leaderboard table presented in the paper includes the version numbers of each model, whereas the platform does not. Including version numbers can allow people to accurately track model performance over time. It would also be helpful to include another metric:  average cost per generation, so people could better interpret results with budget considerations

Some Questions:
- Line 151 - “paper snippets” should be explicitly defined

**Strengths Contributions:**

- The paper is clearly written, I enjoyed reading it
- Evaluating long-form citation-attributed generation is a challenging yet important task, especially with the recent emergence of DeepResearch-like systems. This work introduces an arena-style evaluation framework to this domain, which is novel and insightful
- The evaluation platform is well-designed and easy to use. I tested it on several cases and found it could be a useful tool for tasks such as finding some papers. The interface also displays all retrieved papers, which makes the generation process transparent and easy to follow. I believe the platform can attract many users to contribute voting data
- The collected voting data covers a wide range of question categories and subjects. I noticed that the platform's leaderboard also includes latest models, such as MiniMax-M1 and Claude-4-*, and more voting data after the paper submission. This is a positive development
- The results and analyses of different LLMs show important limitations to address
- The proposed meta-evaluation benchmark in the last section is also valuable, and can be used for future work to evaluate and develop LLM-as-Judge systems in the domain

---

> ### Author Rebuttal · Authors · 2025-07-31
>
> Thank you for taking the time to review our paper and provide valuable feedback. We appreciate your recognition of the value of our work. We would like to address your concerns and questions as follows:
>
> &nbsp;
>
> > **W1**: In Section 5.2, the work analyzes response length bias in evaluation and shows that this bias is smaller compared to previous arena-style platforms. However, for literature review generation, citation-related features like the number of citations within response, can also bias user preference. Analyzing such features could be insightful
> >
>
> Thank you for your insightful suggestion. We have added a new analysis to investigate how citation-related features affect user preferences in SciArena. Specifically, we examine two key aspects: (1) the number of citations within a response, and (2) attribution of inline citations to the generated content.
>
> **Analysis of Citation Count.** We investigate the influence of citation count on user preferences within the SciArena framework. Our analysis yields a modest but positive Bradley-Terry coefficient for citation count ($\gamma = 0.039$). In comparison, contemporary work, Search Arena [1], reports a substantially higher coefficient ($\gamma = 0.209$), indicating a stronger user bias toward citation-rich outputs. These findings suggest that while citation count has some influence, it is not a dominant factor in shaping preferences in the SciArena evaluation setting.
>
> **Analysis of Citation Attribution.** We further examine how the correctness of citation-to-claim attribution impacts user preferences. Using the o4-mini model, we classify each citation-response pair into two categories: (1) *supporting*, where the citation backs the response, and (2) *irrelevant or contradicting*, where the citation does not substantiate the claim or directly contradicts it. We observe a statistically positive coefficient for supporting citations ($\gamma = 0.155$) and a negative coefficient for irrelevant or contradicting ones ($\gamma = -0.154$).
> These findings underscore a difference between SciArena and general-purpose retrieval settings. Specifically, the Search Arena paper reports $\gamma_{\text{support}} = 0.29$, $\gamma_{\text{irrelevant}} = 0.27$, suggesting users do not differentiate between supporting and irrelevant citation. They tend to prefer responses with more citations regardless of attribution quality. In contrast, user behavior in SciArena indicates a clear preference for citations that are highly relevant and correctly attributed to the response content.
>
> &nbsp;
>
> [1] Miroyan, Mihran, et al. "Search Arena: Analyzing Search-Augmented LLMs." *arXiv preprint arXiv:2506.05334* (2025).
>
> &nbsp;
>
> > **W2**: The leaderboard table presented in the paper includes the version numbers of each model, whereas the platform does not. Including version numbers can allow people to accurately track model performance over time. It would also be helpful to include another metric: average cost per generation, so people could better interpret results with budget considerations
> >
>
> Thank you for your valuable feedback. We have updated our leaderboard to include the model version numbers and the average cost per generation for each model.
>
> &nbsp;
>
> > **C1**: Line 151 - “paper snippets” should be explicitly defined
> >
>
> Thank you for raising this point. Paper snippets are concise passages (up to about 480 tokens, roughly one paragraph) that the Semantic Scholar snippet‑search endpoint automatically extracts from a paper’s title, abstract, or main body. They supply the immediate textual context in which a query term appears, allowing relevance to be judged without downloading the full PDF. Multiple snippets are indexed for every paper, and using snippet search lets researchers quickly surface the most pertinent sections of the literature. We have included the definition in the revised manuscript.

---

> > ### Comment · Reviewer_Vq6h · 2025-08-05
> > **Responses**
> >
> > Thank you for the additional analysis and detailed explanation. It clarified my points, and I raised my score now.

---

### Official Review · Reviewer_Yny7 · 2025-07-03

**Rating:** 5
**Confidence:** 4

**Summary:**

This work introduces a platform called SciArena used for evaluating foundation models by researchers through a voting system.  A leaderboard of those models is also provided using the Elo rating. Moreover, a benchmark called SciArena-Eval is introduced for assessing model-based evaluators.

**Dataset Code Accessibility:**

Yes

**Dataset Code Comments:**

I was able to access and review the data.

**Ethical Considerations:**

No, there are no or only very minor ethics concerns

**Final Justification:**

The authors have addressed my concerns and I am raising my score.

**Limitations Weaknesses:**

- The fact that the evaluation of the models is based mainly on votes means that there is a lack of an objective metric that can systematically measure the accuracy of the models.
- Maintaining a voting-based platform requires the continuous engagement of the research community, which might become less and less active with time.
- Open voting might entail the risk of malicious users exploiting the platform to get more votes for certain models.
- A few typos have been observed:
- line 30: relies -> rely
- line 156: addressed -> addresses

**Strengths Contributions:**

- The paper is well-written and clearly explained in general.
- This work provides an important scientific contribution to evaluating foundation models.
- Research experts’ votes have been used to evaluate the models
- Extensive results have been presented to support the authors’ claims.

---

> ### Author Rebuttal · Authors · 2025-07-31
>
> Thank you for taking the time to review our paper and provide valuable feedback. We appreciate your recognition of the value of our work. We would like to address your concerns and questions as follows:
>
> &nbsp;
>
> > **W1:** The fact that the evaluation of the models is based mainly on votes means that there is a lack of an objective metric that can systematically measure the accuracy of the models.
> >
>
> Thank you for the thoughtful comments.
>
> Our study focuses on open-ended, literature-grounded generation tasks. In such settings, unlike clearly verifiable tasks such as mathematical problem solving, it is inherently difficult to define fully objective evaluation metrics. This challenge is well recognized in general-purpose instruction-following benchmarks such as MT-Bench, WildBench, and LLM-Bar, where carefully curated human preferences are treated as the gold standard.
>
> This challenge is a key motivation for our focus on community-driven and human expert evaluation. Our aim is to provide high-quality, domain-expert preference data that offers a reliable and consistent benchmark for evaluating LLMs in complex scientific contexts.
>
> We acknowledge the existence of alternative approaches: (1) Using LLM-based evaluators guided by pre-specified criteria. (2) Creating detailed, expert-annotated rubrics for each example, as in [1]. However, both approaches depend on subjective design decisions and ultimately require some grounding in a gold standard of human judgment.
>
> In our paper, we demonstrate strong inter-annotator agreement and annotator self-consistency in our initial data collection. Additionally, we show that votes collected from the broader community are of high quality (see our response to W3). These findings support the reliability of our human preference data as a practical and consistent evaluation benchmark for LLMs on scientific literature tasks.
>
> &nbsp;
>
> [1] Ruan, Jie, et al. "ExpertLongBench: Benchmarking Language Models on Expert-Level Long-Form Generation Tasks with Structured Checklists." 2025
>
> &nbsp;
>
> > **W2:** Maintaining a voting-based platform requires the continuous engagement of the research community, which might become less and less active with time.
> >
>
> Thank you for your thoughtful comment. We fully agree that sustained engagement from the research community is crucial for the long-term success of the SciArena platform. To achieve this goal, we carefully designed the SciArena platform to enhance the quality of scientific literature retrieval and improve response generation efficiency, directly addressing the limitations of existing platforms (Chatbot platforms with web search enabled, and DeepResearch platform) for scientific literature tasks.
>
> As discussed in Section 3.2, our user study compares two major types of platforms used for scientific literature tasks:
>
> 1. Chatbot platforms with search capabilities, including ChatGPT GPT-4o (Web Search Enabled) and Perplexity AI. Our user study reveals that they often fail to provide relevant citations and sometimes reference unreliable sources, including news articles or social media.
> 2. DeepResearch platforms, including ChatGPT DeepResearch and Gemini DeepResearch
> Our study identifies key pain points in both categories. Users report that these platforms typically take more than five minutes to generate responses, significantly impacting usability.
>
> **To address the issue of failure to retrieve relevant literature,** as described in Section 3.2, we apply the multi-stage retrieval system from ScholarQA, which integrates with the Semantic Scholar API and accesses a large, continuously updated scholarly corpus with both abstract and paper snippet index. To quantitatively evaluate the effectiveness of our retrieval pipeline, we conduct an additional analysis. We randomly sample 100 queries and, for each query, randomly sample 3 out of the 30 retrieved paper passages, resulting in 300 query–retrieved content pairs. Three of the authors serve as evaluators to assess the relevance of each pair. For 47 of these pairs, the evaluators are unable to make a relevance judgment within a 3-minute review due to unfamiliarity with the content. Among the remaining 253 examples, 221 (87.4%) are judged as relevant, and 32 (12.6%) as irrelevant, demonstrating the overall reliability of our retrieval module.
>
> **To improve the efficiency of returning responses after users submit queries**, we have carefully optimized each stage of the generation pipeline (separate immediate steps include query moderation, multi-stage literature retrieval, response generation, and post-processing for unified formatting).Our engineering efforts focus on implementing asynchronous query handling, minimizing redundant API calls, and caching frequently accessed metadata to reduce load on the retrieval pipeline. As a result, the average latency from query submission to LLM response is reduced to 28.7 seconds, significantly enhancing user experience without compromising the depth or quality of the response.
>
> **To promote long-term user engagement,** we have implemented a beta version of a personalized feedback system. Every time a user reaches a new milestone of 50 votes, they receive an update that includes: (1) A summary of their voting activity; (2) Personalized insights based on their voting patterns; (3) Tailored suggestions for tasks or domains where certain models excel, based on their own voting history. This system is designed to both recognize users' contributions and provide meaningful feedback, thereby encouraging continued participation and fostering a sense of community.
>
> As stated in the paper, we remain committed to actively maintaining and expanding the SciArena platform by continuously integrating newly released models. Since the submission, we have added ten new LLMs, including the Claude-4 series, Grok-4, and Minimax-M1. We have also been actively posting regular updates on social media platforms such as Twitter, LinkedIn, and Discord. **With these efforts, we have collected over 7,000 new votes since the paper submission, significantly expanding the original set of 8,101 votes.**
>
> &nbsp;
>
> > **W3**: Open voting might entail the risk of malicious users exploiting the platform to get more votes for certain models.
> >
>
> We acknowledge the concern regarding potential exploitation of the platform through open voting. To address this risk, we have implemented protective measures including:
>
> 1. The use of Cloudflare to detect and block suspicious traffic patterns.
> 2. The use of CAPTCHA verification to prevent automated voting by bots.
> 3. The same algorithm as Chatbot Arena for detecting anomalous users.
> 4. The content moderation check ensures the query is not potentially harmful and is related to scientific literature tasks.
> 5. The model names are hidden until the users submit the votes.
> 6. The response post-processing step, as described in Section 3.2, unifies the format of model responses. This helps mitigate potential evaluation biases introduced by response formatting and makes distinctive model patterns more difficult to detect.
>
> Since the paper submission, we have collected over 7,000 additional votes to date. To validate the quality of the new collected votes, we randomly sampled 150 examples from the newly collected data and assigned 50 examples to each of three authors for manual review. For 17 of these, the reviewers were unable to make a relevance judgment within 10 minutes due to unfamiliarity with the content. For the remaining 133 examples, each was labeled according to the following categories:
>
> 1. Clearly reasonable vote: The vote is clear and would be widely agreed upon
> 2. Vague vote: The selected vote is acceptable, but alternative votes could also be justified
> 3. Wrong vote: The voting outcome is likely to be disagreed with by most reviewers.
>
> We get the following results:
>
> |  | Clearly Reasonable | Vague | Wrong |
> | --- | --- | --- | --- |
> | Count | 98 (73.7%) | 23 (17.3%) | 12 (9.0%) |
>
> We found that approximately 73.7% of votes are clearly reasonable, while 17.3% are acceptable but somewhat ambiguous. Only 9.0% were judged as incorrect. These results suggest that our preference data is reliable.
>
> &nbsp;
>
> > **W4**: A few typos have been observed: line 30: relies -> rely; line 156: addressed -> addresses
> >
>
> Thank you for your careful review. We have corrected the identified typos and conducted another round of thorough proofreading of the paper.

---

> > ### Comment · Reviewer_Yny7 · 2025-08-07
> >
> > I have read the rebuttal response.
> > The authors have addressed my concerns and I am raising my score.

---

> ### Author Response · Authors · 2025-08-06
> **Kind Request for Any Remaining Comments Before Discussion Ends**
>
> Dear Reviewer,
>
> We sincerely appreciate your thoughtful feedback and constructive comments throughout the review process. As the author–reviewer discussion comes to a close, we would be grateful if you could share any remaining concerns or points you'd like to discuss. We understand this is a busy time, and we truly value the time and effort you've dedicated to reviewing our work and engaging with our responses.
>
> Thank you!
>
> The Authors

---

### Official Review · Reviewer_kkAU · 2025-07-03

**Rating:** 5
**Confidence:** 4

**Summary:**

The paper contributes SciArena, which is an open evaluation platform for scientific literature summary and grounding using foundation models. On top of data, code and benchmark, the authors have also provided the access point to their platform, which works quite well based on some tests I‘ve run, especially in tasks where ChatGPT [deepsearch] has difficulties. This is a great example of the science community in combining expert-level human intelligence with SOTA foundation models.

**Dataset Code Accessibility:**

Yes

**Ethical Considerations:**

No, there are no or only very minor ethics concerns

**Final Justification:**

I appreciate your efforts to improve the work. I maintain my score of accept and look forward to the revised version.

**Limitations Weaknesses:**

- Double check how „Chatbot Arena“ is written. I have seen different forms in the text, e.g.,  „ChatbotArena“ and „Chatbot Arena“.
- L170: „For hybrid reasoning models, we enable their „thinking“ mode“. Why?
- What are the difference between „chatbot platforms“ and „agent-based platforms“? Please add the details to Section 3.2, maybe include both their definitions and implementation requirements.
- I would refrain from using words like „sometimes“, „occasionally“ without providing the numbers. I encourage the authors to provide numbers in appendix, e.g., 4/100 ChatGPT is better than SciArena.
- The explanation of BT model is not ideal. Why setting -1 and and 1 for X_i? I would add the following and give an example.
    - The choice of $+1$ for one model and $-1$ for the other is purely a modeling convention. What matters is the **difference** $\beta_m - \beta_{m'}$, not their positions in the comparison. This difference is mapped through $\sigma$ to model the win probability.
    - This is an example. For a comparison between Model A and Model B:
    \begin{align*}
    X &= [1, -1], \quad Y = 1 \quad \Rightarrow \quad \text{A wins}, \\
    X^\top \beta &= \beta_A - \beta_B, \\
    \sigma(\beta_A - \beta_B) &\approx \text{Probability A wins}.
    \end{align*}
    - Swapping roles yields $X = [-1, 1]$, $Y = 0$ — same information, same gradient, same result.
- I would call the „accuracy“ score used in Section 4.3 „consistency“, which makes it easier for readers to associate the measurement.
- In the first paragraph of Section 5.1, mention that the numbers in the brackets are Elo ratings.
- The word „finer-grained questions“ on L.283, what do the authors mean by that? Give examples and then point the readers to the appendix.
- Table 2: reformatting needed.
    - Why is the last column marked with „Elo Score“? I think all of the last five columns are Elo ratings. What is the last column?
    - The red/blue highlight does not do the justice of reading. I would separate the table into two parts, one for open-source, the other for proprietary. One can add coloring of those two parts but it is not needed. Then highlight the best performers and 2nd best ones within each part.
- L.304: „the response from a top-3 model was voted as worse“. What does it mean?
- Does the users get to see after casting the votes which model performs better, e.g., ChatGPT is chosen over Claude on Q1? This gives something back to the users in that they get some intuitions of which foundation models are suitable for their fields/tasks.
- The histogram in D.2, what does „citation counts“ mean?
- Check the sorting of tables. E.g., the table under D.1 is not properly sorted, which hinders readers understanding. Maybe per field is a good a idea. Then we get an idea of representativeness across disciplines.

**Strengths Contributions:**

- Great efforts of the team and it sets a standard for data&benchmark track. It has been one of the best papers I‘ve reviewed in the past 12 months (out of 15 papers). Kudos!
- The authors have done a good job organizing the paper and allow readers to access extensive information via the appendix.
- The authors have tested a large set of popular SOTA open-source and proprietary models.
- The leaderboard not only benefits the AI research on foundation models but also the scientific community in using those in actual research.

---

> ### Author Rebuttal · Authors · 2025-07-31
>
> Thank you for taking the time to review our paper and provide valuable feedback. We appreciate your recognition of the value of our work. We are particularly impressed by the extensive and high-quality nature of your review, a rarity that we greatly appreciate.
> We would like to address your concerns and questions as follows:
>
> &nbsp;
>
> > **W1**: Double check how "Chatbot Arena" is written. I have seen different forms in the text, e.g., "ChatbotArena" and "Chatbot Arena".
> >
>
> We have revised Line 31 from "ChatBotArena" to "Chatbot Arena" for name consistency across the paper.
>
> &nbsp;
>
> > **W2**: L170: ``For hybrid reasoning models, we enable their "thinking" mode.'' Why?
> >
>
> We enable this mode because the scientific literature tasks demand deep reasoning: the model must interpret users’ scientific queries, comprehend the content of retrieved academic papers, and generate literature-grounded, long-form responses. Disabling this mode could underestimate the model’s true capabilities in scientific literature tasks. We have added a sentence to clarify this rationale.
>
> &nbsp;
>
> > **W3:** What are the difference between "chatbot platforms" and "agent-based platforms"? Please add the details to Section 3.2, maybe include both their definitions and implementation requirements.
> >
>
> In Section 3.2, we compare SciArena with two types of platforms through a user study:
>
> - Chatbot platforms with search capabilities, such as ChatGPT GPT-4o (Web Search Enabled) and Perplexity AI. These systems typically provide immediate responses by retrieving and summarizing relevant web content in real-time, but they do not maintain long-term planning or reasoning across multiple steps.
> - Agent-based platforms, such as ChatGPT DeepResearch and Gemini DeepResearch. These platforms rely on a more complex agentic architecture designed to perform multi-step research and reasoning tasks. They generally require significantly more time (often several minutes) to generate a comprehensive final response, due to iterative task decomposition, tool use, and web interactions.
>
> We have included the definitions in the revised manuscript and renamed "agent-based platform" to "DeepResearch platform" to improve clarity.
>
> &nbsp;
>
> > **W4:** I would refrain from using words like "sometimes", "occasionally" without providing the numbers. I encourage the authors to provide numbers in appendix, e.g., 4/100 ChatGPT is better than SciArena.
> >
>
> We appreciate the reviewer’s feedback regarding our use of terms like “sometimes” and “occasionally”. These terms appear in Section 3.2 and Appendix C.1 in the context of our user study, which initially focused on collecting qualitative feedback without recording precise quantitative counts.
>
> In response to the reviewer’s suggestion, we conducted a follow-up user study aimed at quantifying user preferences. We involved three of the original four participants and asked each to evaluate 10 examples using SciArena, ChatGPT with Search, and ChatGPT DeepResearch. Below are the quantitative results:
>
> *1. Number of times each system was preferred based on overall user experience (by user):*
>
> |  | User 1 | User 2 | User 3 |
> | --- | --- | --- | --- |
> | SciArena | 4 | 6 | 4 |
> | ChatGPT with Web Search | 2 | 1 | 0 |
> | ChatGPT DeepResearch | 4 | 3 | 6 |
> &nbsp;
>
> *1. Proportion of citations users found unreliable (e.g., citing untrusted sources)*
>
> |  | User 1 | User 2 | User 3 |
> | --- | --- | --- | --- |
> | SciArena | 10/176 | 7/158 | 11/147 |
> | ChatGPT with Web Search | 35/92 | 21/85 | 42/90 |
> | ChatGPT DeepResearch | 23/114 | 16/108 | 24/113 |
>
> &nbsp;
>
> > **W5:** The explanation of BT model is not ideal. Why setting -1 and and 1 for X\_i? The choice of +1 for one model and -1 for the other is purely a modeling convention. What matters is the difference , not their positions in the comparison.…
> >
>
> Thanks for raising this point. We agree that the choice of $X_i = (-1, 1)$ in the BT model is a modeling convention, and that was the value we used in our code implementation. As you point out, any antisymmetric encoding that preserves the difference $\beta_B - \beta_A$ is mathematically equivalent. We have clarified this point in the revised manuscript by adding a sentence to explain that alternative codings (e.g., $(1, -1)$ or scaled versions) yield identical likelihoods, gradients, and final results.
>
> &nbsp;
>
> > **W6:** I would call the “accuracy“ score used in Section 4.3 “consistency", which makes it easier for readers to associate the measurement.
> >
>
> We have updated the metric name from "accuracy" to "consistency" in Section 4.3 to improve clarity for readers.
>
> &nbsp;
>
> > **W7:** In the first paragraph of Section 5.1, mention that the numbers in the brackets are Elo ratings.}
> >
>
> We have revised the paragraph to clarify that the numbers in brackets represent Elo ratings.
>
> &nbsp;
>
> > **W8:** The word "finer-grained questions" on L.283, what do the authors mean by that? Give examples and then point the readers to the appendix.
> >
>
> By "finer-grained questions", we refer to the different question categories (e.g., *Conceptual Explanation*, *Methodology Inquiry*) presented in Figure 3 and discussed in Section 4.3. We have revised the sentence to include a reference to Figure 10 in Appendix E.1, which shows the Elo ratings of the top-5 models across these specific question categories.
>
> &nbsp;
>
> > **W9**: Table 2: reformatting needed …
> >
>
> We have clarified in the caption of Table 2 that the last five columns represent Elo ratings. Additionally, we have separated the table into two sections (i.e., proprietary models and open-source models) and highlighted the top-2 performers within each section.
>
> &nbsp;
>
> > **W10**: L.304: "the response from a top-3 model was voted as worse". What does it mean?
> >
>
> We have revised the sentence to clarify the intended meaning. The updated version is (the bold text highlights the revision we made):
>
> "We use the collected human preference data to analyze examples that are especially challenging for current foundation models. Specifically, we filter for instances where (1) users judged both two models' responses to be poor, **because such examples expose limitations shared across multiple systems and reveal systematic weaknesses**; or (2) **the two responses are from one of the top-3 models and one of the other models, and** the response from a top-3 model was voted as worse, **because these cases help us understand the limitations of the best‑performing models.**..."
>
> &nbsp;
>
> > **W11:** Does the users get to see after casting the votes which model performs better, e.g., ChatGPT is chosen over Claude on Q1?
> >
>
> Yes, users can see which model performs better after submitting their votes. The model names are hidden until the vote is submitted.
>
> &nbsp;
>
> > **W12**: The histogram in D.2, what does "citation counts" mean?
> >
>
> We have revised the figure caption to clarify that "citation counts" refers to the number of citations included within a model-generated response.
>
> &nbsp;
>
> > **W13**: Check the sorting of tables. E.g., the table under D.1 is not properly sorted, which hinders readers understanding. Maybe per field is a good idea. Then we get an idea of representativeness across disciplines.
> >
>
> We have revised Table 7 (Biographies of researchers involved in SciArena initial human preference data collection) in Section D.1, sorting it first by "discipline" and then by "field". We have also double-checked all other tables and figures to ensure they are appropriately sorted for readability.

---

> > ### Comment · Reviewer_kkAU · 2025-08-05
> > **Final comments**
> >
> > I have reviewed all the responses provided by the authors, which are quite detailed.
> > I appreciate the authors' efforts to improve the paper's readability and clarity.

---

### Official Review · Reviewer_N6kA · 2025-07-04

**Rating:** 4
**Confidence:** 4

**Summary:**

The paper introduces SciArena, a collaborative platform for evaluating LLMs on scientific literature tasks. SciArena uses the "arena" model popularized by ChatbotArena, collecting preference votes from human researchers who compare side-by-side model responses to their own real-world queries. The primary contribution of the paper is the SciArena platform itself with 15 different LLMs ranked via an Elo rating system based on 8000 human preference votes from a pool of 80 researchers from diverse scientific domains. The second contribution suggested by the Authros is a new benchmark of 2000 selected pairs of LLM responses that can be used to evaluate the ability of LLMs to identify better/worse literature-grounded model responses.

**Additional Feedback:**

This is a well-written paper and the created tool might become more useful, but I have to share my main reservations and the reasons for my less then stellar evaluation. Firstly, I really doubt that in current form the arena is really useful without dedicated "deep research" tools. Yes, a couple of months ago people would upload a bunch of papers retrieved externally into an LLM (using a tool such as NotebookLM or similar) and work from there, but I feel that we have already passed that point and we should focus on agentic tools for scientific writing. In principle, I don't object to arena-style evaluation of LLMs, but using 70 students as expert evaluations is definitely not enough to convince me that this evaluation really ranks the LLMs by reliability and ability to produce well-grounded responses. Let's take our discipline, the Appendix lists just 3 annotators from Computer Science, would I base my choice of tool on the opinions of one graduate and two phd students? Certainly not.
My second reservation concerns the presented benchmark, I really think it has no merit and no value, I would consider the paper for publication in this track without it just based on the tool alone. But my main problem is with the fact that the tool does not seem to do what it promises to do - it basically evaluates the ability of LLMs to ingest and analyze a buch of pdf files. This separation of retrieval and generation is, in my opinion, the biggest weakness of the presented tool.
Of course I appreciate the work put into designing and developing the platform, I understand that this is the type of paper that is hard to judge. My personal opinion is that the scientific contribution of this paper is low and the utility of the tool is limited due to the exclusion of the most important class of models, but I will carefully read the comments of other reviewers and Authors' responses as I remain open to changing my evaluation.

**Dataset Code Accessibility:**

Yes

**Dataset Code Comments:**

The paper provides direct links to the live platform, the dataset on Hugging Face, and the code on GitHub. The dataset is hosted on a standard and highly accessible platform. The code is under a permissive (MIT/CC-BY 4.0) license. The dataset is well-documented within the paper itself, with detailed descriptions of the data collection process, quality control, and statistics. The appendices provide enough details  for reproduction, including the prompts used for generation and post-processing and the specific configurations and versions of all evaluated models.

**Ethical Comments:**

The work does not raise significant ethical concerns that require further review. The authors have addressed the key ethical issues typical in this type of research. Thehe paper claims that the annotators in the initial study were compensated at a rate above the local average wage. Importantly, the authors show a strong awareness of potential biases in arena-style evaluations and they try to mitigate them implementing a response post-processing pipeline to standardize formatting and reduce stylistic biases. I was also pleasantly surprised that the platform incorporates a content moderation check to filter potentially harmful queries, preventing the generation of unsafe content and its inclusion in the dataset.
In one of the experiments the model originating from Allen AI Institute took the first place, and part of the Authors is affiliated with this institution, I am not sure if this might create a conflict of interests or financial consequences, but I feel that the Authors might have commented on that fact.

**Ethical Considerations:**

No, there are no or only very minor ethics concerns

**Final Justification:**

I am changing my final evaluation as the rebuttal slightly reduces my initial concerns, although I remain slightly unconvinced about the usefulness of the presented evaluation arena.

**Limitations Weaknesses:**

The paper has one significant weakness: in fact, it does not evaluate the LLMs used for automatically generate surveys and reports based on scientific papers. The Authors acknowledge this fact, but in a slightly indirect way, trying to wave off the problem by saying that agentic systems also query sources other than scientific literature, such as websites and blogs (and they must be aware that this can be to a large extend limited by a system prompt). Main tools for doing AI-based scientific research (such as OpenAI Deep Research or Gemini 2.5 Deep Research) are excluded from the SciArena because of usage rate limits and problems with integrating their APIs. And so we are left with traditional LLMs that are shown the context of scientific papers coming from the RAG pipeline that is not under the command of these LLMs. Obviously, the quality of the final response is critically dependent on the fixed, upstream retrieval module from ScholarQA. A model might be penalized for a poor response that was actually caused by the retriever failing to provide relevant context. The paper does not analyze or attempt to disentangle the impact of retrieval failures on the final model rankings. This makes it difficult to assess the models' synthesis and reasoning capabilities independently of the retriever's performance.
Also, there is a contradiction in the paper. The Authors explicitly say that SciArena overcomes the problems of static benchmarks, and then they introduce SciArena-Eval benchmark with a fixed set of 2000 static examples. This benchmark will face the same problem of aging as the models it is designed to evaluate improve. The authors do not propose a clear plan or mechanism for continuously updating or versioning SciArena-Eval to keep it relevant.

**Strengths Contributions:**

It is quite obvious that the use of LLMs in scientific research and writing is quickly gaining momentum, so the work is timely and important. I would say that the key strength of the paper is the design of the SciArena application to showcase SOTA in literature-based query answering. I fully agree that we need a tool to evaluate how good LLMs are in providing summaries, performing surveys, and extracting knowledge from scientific literature. I also think that the Authors have put a lot of work in both building the platform and recruiting human annotators. However, I don't think the Authors should use the term "human expert" so frequently as all the annotators are either graduate students or PhD students. Yes, they should be somewhat familiar with the process of literature search, but I seriously doubt if these annotators can fully evaluate the performance of the models. At least, there is significant inter-annotator agreement and self-consistency, the kappas are looking really strong, so maybe my hesitations about the level of expertise of annotators is not warranted.
The paper is very well written and easy to follow. The methodology for the platform, the description of the entire workflow, and analyses of the preference data are thorough. The Authors know about methodological problems with arena-style evaluations and try hard to counter at least some of the problems, which is definitely a strength.
I have to say though that I don't regard the SciArena-Eval benchmark to be a significant contribution, as a matter of fact, I don't think it is a contribution at all. At the peril of being blunt I will say that this benchmark strikes me as extremely artificial and the only reason it is there, seems to me, is to qualify for the datasets & benchmarks track. Because the main contribution of the paper is neither a dataset nor a benchmark, but an interesting application. The SciArena-Eval is a set of triplets with a query and two responses from two LLMs, and the benchmark's purpose is to measure if an LLM can identify the response preferred by the majority of human annotators. What would be the utility of such benchmark? How would anyone use such benchmark in practice? I really see no point in Section 6 other than to have a paper that pretends to showcase a benchmark and not a tool

---

> ### Author Rebuttal · Authors · 2025-07-31
>
> Thank you for taking the time to review our paper and provide valuable feedback. We would like to address your concerns and questions as follows:
>
> &nbsp;
>
> ### W1: Clarifying why we do not prioritize Deep Research systems and use a uniform literature retrieval pipeline
>
> Our goal is to assess the language-reasoning capabilities of frontier LLMs under a fixed, transparent, retrieval-augmented agentic framework. This component-level evaluation is important for two primary reasons:
>
> **1. Scientific control**
>
> By holding the retrieval component of the agentic system constant, we eliminate confounding factors from orchestration heuristics and retrieval quality. This ensures low-variance, reproducible measurements that developers can trust when selecting or comparing LLMs.
>
> **2. Complementarity to end-to-end agentic Deep Research benchmarks**
>
> Agentic systems like Deep Research ultimately depend on the LLMs at their core. SciArena helps determine whether observed performance gains are due to improved agentic orchestration or the intrinsic capabilities of the underlying models.
>
> As part of our recent updates, we have integrated a deep research system (i.e., o4-mini-deep-research-2025-06-26) into SciArena. When this system is selected, it is always paired against o4-mini, enabling direct comparisons between its agentic and non-agentic variants. This setup provides a clearer picture of how much agentic design contributes to performance beyond the base model's inherent reasoning capabilities. A separate results board will be introduced once sufficient user votes have been collected to support statistically robust comparisons.
>
> We plan to add more systems, such as Gemini 2.5 Deep Research and Claude Research, once their APIs are available. We remain committed to actively maintaining and expanding the SciArena platform by continuously integrating newly released models. Since the paper submission, we have added ten latest LLMs and one Deep Research system.
>
> **Regarding the concerns about retrieval quality,** as discussed in the second paragraph of Section 3.1, the SciArena platform is designed with a strong emphasis on the retrieval pipeline, as the quality of retrieved paper contents directly affects the informativeness and accuracy of the final responses. To this end, we adopt the multi-stage retrieval system from ScholarQA. To quantitatively evaluate the effectiveness of our retrieval pipeline, we conduct an additional analysis. We randomly sample 100 queries and, for each query, randomly sample 3 out of the 30 retrieved paper passages, resulting in a total of 300 query–retrieved content pairs. Three of the authors serve as evaluators to assess the relevance of each pair. For 47 of these pairs, the evaluators are unable to make a relevance judgment within a 3-minute review due to unfamiliarity with the content. Among the remaining 253 examples, 221 (87.4%) are judged as relevant, and 32 (12.6%) as irrelevant, demonstrating the overall reliability of our retrieval pipeline.
>
> &nbsp;
>
> ### W2: Explaining necessity of proposed meta-evaluation benchmark, SciArena-Eval
>
> We are happy to clarify the added-value of SciArena-Eval.
>
> Automated evaluation of open-ended LLM outputs is increasingly crucial as model responses grow more complex and domain-specific, particularly in scientific contexts where expert annotation is costly. A common approach to address this challenge is to use LLM-as-a-Judge, as a proxy for human evaluation. However, the reliability of these LLM-based evaluators remains uncertain. In particular, for scientific literature tasks, there is a lack of high-quality meta-evaluation benchmarks (i.e., benchmarks focused on evaluation of evaluation methods).
>
> SciArena-Eval directly addresses this gap by providing a meta-evaluation benchmark with expert-annotated gold standards. This allows researchers and practitioners to assess the capabilities and trustworthiness of LLM-based evaluators by measuring how well their scores align with high-quality human evaluations. The SciArena-Eval meta‑evaluation framework starts with a set of instructions $N$ and a pool of models $M$ and a sampled model pairs producing a pool of response pairs $O$. SciArena already collects human votes for instances in $O$. The subset $S$ of instances from $O$ are chosen as a fixed meta-evaluation set. A large pool of automated evaluators $E$ score each pair from $S$. Their output judgments are then compared with human preferences. Note that in theory it is possible to perform meta-evaluation on the full set of votes from SciArena, but running a large set of LLM evaluators is prohibitively expensive, and the results won't be reproducible due to dynamic nature of the platform. As such, the standard method is to opt for a smaller fixed subset.
>
> In practice, SciArena-Eval enables three key use cases:
>
> - **Benchmarking**: Developers can test new LLM evaluators on scientific tasks by comparing their outputs against human-annotated references
> - **Model analysis**: By analyzing correlations at both the instance and system level, researchers can study the behavior of LLM-based evaluators and uncover when and how they might fail
> - **LLM-based evaluator improvement**: Insights from the benchmark can guide the development of more reliable LLM-based evaluators tuned for scientific literature task
>
> > The Authors explicitly say that SciArena overcomes the problems of static benchmarks, and then they introduce SciArena-Eval benchmark with a fixed set of 2000 static examples. This benchmark will face the same problem of aging as the models it is designed to evaluate improve.
> >
>
> Thank you for raising this point; the roles of SciArena and SciArena‑Eval differ:
>
> SciArena is a live benchmarking platform designed to evaluate the performance of LLMs on scientific literature tasks. It continuously evolves by incorporating new models and user-submitted questions as they become available. This dynamic nature ensures that researchers and practitioners can gain up-to-date insight into how new LLM releases perform under identical retrieval conditions, enabling timely model selection and debugging.
>
> SciArena-Eval serves a different purpose: it is intended as a fixed, gold-standard benchmark to evaluate the quality of LLM-based evaluators as proxy for human judgments. LLM-based evaluators or judges are commonly used in both academia and industry and SciArena-Eval allows evaluating their quality against scientific tasks. As mentioned above, freezing the 2,000 expert-labeled examples is a design decision to ensure reproducibility, enable longitudinal analysis, and amortize the substantial cost of expert annotation.
>
> We acknowledge the eventual limitations of a fixed dataset. To address this, we plan to periodically release updated versions (e.g., v2, v3) based on collected data from SciArena. These updates will ensure that SciArena-Eval remains a relevant and reliable evaluation resource over time.
>
> &nbsp;
>
> ### W3: Clarifying reliability of collected human preference data
>
> > …using 70 students as expert evaluations is definitely not enough to convince me that this evaluation really ranks the LLMs by reliability and ability to produce well-grounded responses.... At least, there is significant inter-annotator agreement and self-consistency, the kappas are looking really strong, so maybe my hesitations about the level of expertise of annotators is not warranted.
> >
>
> Thank you for raising this point.
>
> While the annotators for the evaluation are graduate and PhD students, we ensure a strong baseline of expertise during recruitment: each has at least two peer-reviewed publications, while 54 out of 77 have more than five (as detailed in Appendix D.1). As you note, the inter-annotator agreement is high (Section 4.3), which supports the reliability of their assessments.
>
> Since the public release of our platform, we have collected over 7,000 additional votes. To validate the quality of the new collected votes, we randomly sampled 150 examples from the newly collected data and assigned 50 examples to each of three authors for manual review. For 17 of these, the reviewers were unable to make a relevance judgment within 10 minutes due to unfamiliarity with the content. For the remaining 133 examples, each was labeled according to the following categories:
>
> 1. Clearly reasonable vote: The vote is clear and would be widely agreed upon
> 2. Vague vote: The selected vote is acceptable, but alternative votes could also be justified
> 3. Wrong vote: The voting outcome is likely to be disagreed with by most reviewers.
>
> We get the following results, which suggests that the preference data collected from community is reliable.
>
> |  | Clearly Reasonable | Vague | Wrong |
> | --- | --- | --- | --- |
> | Count | 98 (73.7%) | 23 (17.3%) | 12 (9.0%) |
>
> &nbsp;
>
> ### E1: Clarifying that we do not evaluate AI2's models in our work
>
> > In one of the experiments the model originating from Allen AI Institute took the first place, and part of the Authors is affiliated with this institution … I feel that the Authors might have commented on that fact.
> >
>
> We do not include models from the Allen Institute for AI (AI2) in the SciArena platform or this paper. The best publicly available model from AI2, OLMo-2, supports a maximum context window of only 4,096 tokens. Our platform focuses on evaluating models with context windows larger than 16K tokens, as we consider this a minimum requirement for literature-based tasks, which typically require reasoning across multiple scientific papers.
>
> The top-performing models reported in our paper are Gemini-2.5-Pro (Google), o4-mini (OpenAI), Grok-3 (xAI), GPT-4.1 (OpenAI), DeepSeek-V3 (DeepSeek), and DeepSeek-R1 (DeepSeek).
>
> We actively maintain and expand the platform to include newly released models. If future AI2 models meet our inclusion criteria, we will consider evaluating them. In such cases, we will explicitly disclose any potential conflicts of interest.

---

> ### Author Response · Authors · 2025-08-06
> **Kind Request for Any Remaining Comments Before Discussion Ends**
>
> Dear Reviewer,
>
> We sincerely appreciate your thoughtful feedback and constructive comments throughout the review process. As the author–reviewer discussion comes to a close, we would be grateful if you could share any remaining concerns or points you'd like to discuss. We understand this is a busy time, and we truly value the time and effort you've dedicated to reviewing our work and engaging with our responses.
>
> Thank you!
>
> The Authors

---

### Author Response · Authors · 2025-08-05
**Kind Request for Response to Our Rebuttal**

Dear Reviewers,

Thank you again for the thoughtful feedback you shared on our submission. We have incorporated your suggestions and posted a detailed response. As the discussion phase ends in two days, we would be grateful if you could take a moment to look over our rebuttal and let us know whether any points need further clarification.

We appreciate your time and support.

Best regards,

The authors

---

### Note · Authors · 2025-08-15

We sincerely thank all reviewers for their insightful comments and constructive feedback. We greatly appreciate the recognition of our work’s contributions, including:
- Clear and well-written presentation (all four reviewers). *“It has been one of the best papers I‘ve reviewed in the past 12 months (out of 15 papers)”* (*kkAU*)
- Addressing an important scientific contribution to evaluating foundation models in scientific literature tasks (*kkAU, Vq6h, Yny7*)
- The design and usability of the SciArena platform, which reviewers found intuitive and effective: *“It works quite well based on some tests I’ve run, especially in tasks where ChatGPT [deepsearch] has difficulties”* (*kkAU*); *“I believe the platform can attract many users to contribute voting data”* (*Vq6h*)
- High quality of the collected human voting data (*kkAU, Vq6h, Yny7*)
- Comprehensive evaluation offering insights into current LLM capabilities (*kkAU, Vq6h, Yny7*)
- Introduction of a meta-evaluation benchmark for advancing LLM-as-Judge systems in this domain (*Vq6h*)
---
Our rebuttal addressed the main concerns as follows:
- Reviewer N6kA
    - Clarified that the uniform literature retrieval pipeline serves as a scientific control and complements end-to-end agentic Deep Research benchmarks. Our user study validated the high quality of this pipeline
    - Explained the importance of the proposed meta-evaluation benchmark, SciArena-Eval, for advancing LLM-as-Judge systems in scientific literature tasks
    - Addressed ethical concerns by clarifying that we did not evaluate any models from AI2
- Reviewer kkAU
    - Incorporated presentation improvements based on the reviewer’s detailed  suggestions
- Reviewer Yny7
    - Clarified that the challenges of open-ended, literature-grounded generation tasks motivate our focus on community-driven and expert human evaluation
    - Discussed our efforts to ensure both (1) high-quality votes from the community and (2) sustained research community engagement
- Reviewer Vq6h
    - Added a new analysis on how citation-related features influence user preferences in SciArena
    - Included model version and cost-per-query information on the leaderboard
---
By introducing SciArena and SciArena-Eval, we aim to make a meaningful contribution to advancing LLM research for scientific literature tasks. We are committed to actively maintaining and expanding the SciArena platform by continuously integrating newly released models and ensuring its long-term growth.

---

### Decision · Program_Chairs · 2025-09-18

**Decision:**

Accept (spotlight)

**Comment:**

The paper introduces SciArena, a platform for the online evaluation of LLMs, very similar to LM Arena but tailored specifically to scientific literature tasks. The online leaderboard is complemented by SciArena-Eval, an offline benchmark for *judges* on such tasks.

Most reviewers were highly positive about this work, especially after the author rebuttal that addressed the large majority of concerns and questions. Only reviewer N6kA remains somewhat mixed (but still positive), with concerns regarding the relevance of this work especially in the context of Deep Research agents. Although I believe this is a valid concern, I still see value in its current state as many scientific questions don't necessarily require the depth of such agentic systems.

Given the consensus towards acceptance, and the potential usefulness of this work to the research community (the tool itself may prove quite useful to students and researchers), I am happy to recommend this submission for acceptance -- especially as more visibility should increase its popularity and thus its overall usefulness.